# A Uniform generalization bound for Generative Adversarial Networks

## Abstract

This paper focuses on the theoretical investigation of unsupervised generalization theory of generative adversarial networks (GANs). We first formulate a more reasonable definition of general error and generalization bounds for GANs. On top of that, we establish a bound for generalization error with a fixed generator in a general weight normalization context. Then, we obtain a width-independent bound by applying $\ell_{p,q}$ and spectral norm weight normalization. To better understand the unsupervised model, GANs, we establish the generalization bound, which uniformly holds with respect to the choice of generators. Hence, we can explain how the complexity of discriminators and generators contribute to generalization error. For $\ell_{p,q}$ and spectral weight normalization, we provide explicit guidance on how to design parameters to train robust generators. Our numerical simulations also verify that our generalization bound is reasonable.

## 1 Introduction

The generative adversarial network (GAN) (Goodfellow et al., 2014) is one of most powerful generative models for modeling complex high-dimensional tasks, such as image generation, dialogue generation, and image impainting. Many variants of GANs (Ho & Ermon, 2016; Abadi & Andersen, 2016; Goodfellow et al., 2014; Li et al., 2017; Yu et al., 2018) have also been introduced to reinforce the stability of training processes to obtain more realistic models.

A GAN consists of two neural networks: a discriminator and a generator. Literally, the generator generates simulated data, while the discriminator tries to discriminate between simulated data and real data. The training process of GANs is tantamount to a two-player game between a generator and a discriminator. The main goal is to obtain a good generator, which is able to successfully approximate the distribution of real data. We denote the distribution of real data and the generator-induced distribution by $\mathcal{D}_{real}$ and $\mathcal{D}_g$, respectively. Our goal is to find a generator such that $\mathcal{D}_{real} = \mathcal{D}_g$. We revise the goal as $d(\mathcal{D}_{real}, \mathcal{D}_g) = 0$, with a distribution distance $d(\cdot, \cdot)$. The Jensen-Shannon (JS) divergence is implicitly used in Vanillar GANs (Goodfellow et al., 2014), and the 1-Wasserstein distance is employed in WGANs (Arjovsky et al., 2017). Empirical experiments suggest that the Wasserstein distance is a more sensible measure for differentiating probability measures supported in low-dimensional manifolds.

The generalization properties of GANs are less explored in the literature, and some exceptions are these works(Jiang et al., 2019; Arora et al., 2017; Bartlett et al., 2017; Zhang et al., 2017). Motivated by the supervised learning context, where we say training to be generalized if the gap between the training loss and the test loss is small, we can define the generalization for GANs in a similar way. Concretely, generalization for GANs means that, the population distance between $\mathcal{D}_{real}$ and $\mathcal{D}_g$ is closed to the empirical distance between the empirical distributions of $\mathcal{D}_{real}$ and $\mathcal{D}_g$. Hence, we define the gap between the population and empirical distance as the generalization error. Though our ultimate goal is to minimize the former distance, the latter one is what we minimize in practice. Given that our training process provides a small distance between empirical distributions, a small generalization error indicates that the population distance is also small. In other words, a small generalization error guarantees that the generated distribution is close to the real data distribution.

In fact, the training process of GANs is sample-dependent. In other words, the generator depends on the training data sets, which are random samples from $\mathcal{D}_{real}$. The training process minimizes

$d(\hat{\mathcal{D}}_{real}, \mathcal{D}_g)$, where $\hat{\mathcal{D}}_{real}$ denotes the empirical distribution over samples. The deviation between $\hat{\mathcal{D}}_{real}$ and $\mathcal{D}_{real}$ leads to the generalization error, i.e., the gap between $d(\mathcal{D}_{real}, \mathcal{D}_g)$ and $d(\hat{\mathcal{D}}_{real}, \mathcal{D}_g)$. This motivates us to establish a bound for generalization error, that is, the generalization bound. A tight generalization bound guarantees that the generalization error is small. The highlights and main contributions of this article are summarized as follows:

- We formulate new definitions for both generalization error and generalization bound, which are more reasonable than the definitions in previous work (Arora et al., 2017; Jiang et al., 2019).

- We establish the generalization error bound in a general version, with a *fixed* generator. By applying $\ell_{p,q}$ weight normalization, we obtain a tighter bound.

- We establish the generalization error bound, which *uniformly* holds over any choice of generator. Hence, we can explain how the complexity of the generator class and discriminator class contribute to the generalization error.

- Numerical experiments on Gaussian Mixture models verify that the theory of generalization error bound is consistent with the numerical performance.

## 1.1 RELATED WORK

Some previous works provide theoretical investigations of the generalization of GANs. Arora et al. (2017) introduces a new metric for distributions called $\mathcal{F}$-distance, and defines the generalization for GANs based on this distance. On top of that, the paper shows that generalization does happen with a moderate number of training examples (i.e., when the generator wins, the two distributions must be close in $\mathcal{F}$-distance). However, they analyze the generalization with a fixed generator. Hence, the result is not guaranteed to hold uniformly across all generators. In this paper, we establish generalization error bounds for both scenarios. When a generator is fixed, we provide a tighter bound for generalization error than that in Arora et al. (2017).

Jiang et al. (2019) also established a bound for generalization error using spectral normalization of GANs with a fixed generator. They show the advantages of spectrum control for generalization by constraining discriminator class. By adopting the Rademacher Complexity, Bartlett et al. (2017) yielded a bound of order $O(\sqrt{d^3 k/m})$, where $d, k, m$ stand for the largest discriminator width, the discriminator depth, and the training data size, respectively. Jiang et al. (2019) derived a bound of order $O(\sqrt{d^2 k/m})$. In our work, we establish a bound of order $O(d_1^{\frac{1}{p^*}}\sqrt{k/m})$ with $1/p^* \leqslant 1$, $d_1 \leqslant d$, which is tighter than those from previous works. Moreover, we provide a more general version of the bound for generalization error with a fixed generator, and the result in Jiang et al. (2019) is a special case under the spectral weight normalization. To the best of our knowledge, we are the first to establish a generalization bound for GANs that holds uniformly across all generators.

## 2 PRELIMINARIES

We first introduce the formulation of the generalization bound for GANs. We use $\mathcal{D}_{real}$ for the real data distribution over $\mathbb{R}^{d_0}$, and $\mathcal{N}_d(\mathbf{0}, \mathbf{I})$ for a $d$-dimensional standard Gaussian distribution. We define the sample set as $\mathcal{S} \triangleq \{\mathbf{x}_i\}_{i=1}^m$, where $\{\mathbf{x}_i\}_{i=1}^m$ are i.i.d samples from $\mathcal{D}_{real}$ and denote $\mathbf{z} \sim \mathcal{N}_{l_0}(\mathbf{0}, \mathbf{I})$ as Gaussian noise. We denote the $\ell_{p,q}$-norm of a matrix $\mathbf{A}$ as $\|\mathbf{A}\|_{p,q} \triangleq (\sum_j (\sum_i \mathbf{A}_{i,j}^p)^{\frac{q}{p}})^{\frac{1}{q}}$ and the spectral norm as $\|\mathbf{A}\|_2$ . We denote the conjugate of $p$ as $p*$, with $1/p^* + 1/p = 1$. For an arbitrary distribution $\mu$, $\hat{\mu}$ denotes the empirical distribution over a random sample of size $m$ from $\mu$.

We let $\mathcal{F} = \{f \mid f : \mathbb{R}^{d_0} \to [-1, 1]\}$ denote the function class of discriminators, and $\mathcal{G} = \{g \mid g : \mathbb{R}^{l_0} \to \mathbb{R}^{d_0}\}$ denote the function class of generators. Every generator $g \in \mathcal{G}$ induces a distribution $\mathcal{D}_g$: by applying $g$ to a random sample $\mathbf{z} \sim \mathcal{N}_{l_0}(\mathbf{0}, \mathbf{I})$, then we generate a sample $g(\mathbf{z})$ from $\mathcal{D}_g$. In the context of GANs, both $\mathcal{F}$ and $\mathcal{G}$ are neural network function classes. Specifically, $\forall f \in \mathcal{F}$, $\mathbf{x} \in \mathbb{R}^{d_0}$, $f(\mathbf{x}) = T_{f,k_1+1} \circ \sigma \circ T_{f,k_1} \circ \cdots \circ \sigma \circ T_{f,1} \circ \mathbf{x}$, where $\forall 1 \leq i \leq k_1+1, T_{f,i}(\mathbf{u}) \triangleq \mathbf{W}_{f,i}^\top \mathbf{u} + \mathbf{b}_{f,i}$, and $\sigma(\cdot)$ is a $\rho$-Lipschitz active function. Note that, $\mathbf{b}_{f,i} \in \mathbb{R}^{d_i \times 1}, \mathbf{W}_{f,i} \in \mathbb{R}^{d_{i-1} \times d_i}$, where $d_i$ is the width of the $i$th layer of $f$, and $k_1 + 1$ is the depth of $f$. For convenience, we introduce

$\mathbf{M}_{f,i} \triangleq (\mathbf{b}_{f,i}, \mathbf{W}_{f,i}^\top)^\top$. Similarly, we define the generator class as $\mathcal{G} \triangleq \{g \mid g = T_{g,k_2+1} \circ \sigma \circ T_{g,k_2} \circ \cdots \circ \sigma \circ T_{g,1} \circ \mathbf{z}, \ \mathbf{z} \in \mathbb{R}^{l_0}\}$. Here, $T_{g,i}(\mathbf{u}) \triangleq \mathbf{W}_{g,i}^\top \mathbf{u} + \mathbf{b}_{g,i}$, $1 \leq i \leq k_2 + 1$. Note that, $\mathbf{b}_{g,i} \in \mathbb{R}^{l_i \times 1}, \mathbf{W}_{g,i} \in \mathbb{R}^{l_{i-1} \times l_i}$, where $l_i$ is the width of the $i$th layer of $f$, and $k_2 + 1$ is the depth of $g$. For neural network functions $f \in \mathcal{F}$ and $g \in \mathcal{G}$, we parameterize them as $f_\mathbf{w}, g_\mathbf{v}$, respectively, where $\mathbf{w}, \mathbf{v}$ are the weight parameters. We denote $\mathcal{W}$ and $\mathcal{V}$ as the parameter space of $\mathcal{F}, \mathcal{G}$, respectively. We denote the Lipschitz constant (with respect to the input $\mathbf{x} \in \mathbb{R}^{d_0}$) of $f$ as $L$.

**Weight normalization**. Weight normalization is an efficient regularization method for training robust models. We introduce the $\ell_{p,q}$ and spectral weight normalizations, and establish the generalization theory for such weight normalized neural networks.

Assume that, $\mathcal{F}$ is a neural network function class, which is parameterized as $\mathcal{F} = \{f_\mathbf{w} | \mathbf{w} = (\mathbf{M}_{f,k_1+1}, \ldots, \mathbf{M}_{f,1}), \mathbf{w} \in \mathcal{W}\}$. We define $\mathcal{F}$ with $\ell_{p,q}$ weight normalization as $\|\mathbf{M}_{f,i}\|_{p,q} \leqslant c_{\mathcal{F},i}, i = 1, \ldots, k_1 + 1$. In this context, we define the parameter norm as $\|\mathbf{w} - \mathbf{w}'\|_{p,q} \triangleq \sum_{i=1}^{k_1+1} \|\mathbf{M}_{f,i} - \mathbf{M}'_{f,i}\|_{p,q}/c_{\mathcal{F},i}$.

For spectral weight normalization, we repeat the definition in Jiang et al. (2019)'s work. Assume that, $\mathcal{F}$ is a neural network function class without bias terms. It is parameterized as $\mathcal{F} = \{f_\mathbf{w} | \mathbf{w} = (\mathbf{W}_{f,k_1+1}, \ldots, \mathbf{W}_{f,1}), \mathbf{w} \in \mathcal{W}\}$. We define $\mathcal{F}$ with spectral weight normalization as $\|\mathbf{W}_{f,i}\|_2 \leqslant B_{\mathcal{F},i}, i = 1, \ldots, k_1 + 1$. In this context, we define the parameter norm as $\|\mathbf{w} - \mathbf{w}'\|_2 \triangleq \sum_{i=1}^{k_1+1} \|\mathbf{W}_{f,i} - \mathbf{W}'_{f,i}\|_2/B_{\mathcal{F},i}$.

For $\mathcal{F}$ and $\mathcal{G}$, the parameter norm $\|\cdot\|$ induces the metric parameter space $(\mathcal{W}, \|\cdot\|)$ and $(\mathcal{V}, \|\cdot\|)$, respectively. Hence, we define the Lipschitz constants of $f \in \mathcal{F}$ and $g \in \mathcal{G}$ (with respect to $\|\cdot\|$) as $L_f$ and $L_\mathcal{G}$, respectively.

**GANs**. According to the training process of GANs, we formulate the objective functions as:

$$\min_{g \in \mathcal{G}} \max_{f \in \mathcal{F}} \mathbb{E}_{\mathbf{x} \sim \mathcal{D}_{real}} [\phi(f(\mathbf{x})] - \mathbb{E}_{\mathbf{x} \sim \mathcal{D}_g} [\phi(f(\mathbf{x}))],$$

where $\phi(\cdot) : [-1, 1] \to \mathbb{R}$ is a monotone $L_\phi$-Lipschitz continuous function. The objective function shows that, the discriminator $f$ should give high values to $\mathbf{x} \sim \mathcal{D}_{real}$ and low values to $\mathbf{x} \sim \mathcal{D}_g$. When $\mathcal{D}_{real}, \mathcal{D}_g$ are the same, $f$ is expected to output 0.

The training process for GAN is tantamount to minimizing a specific distance, between $\mathcal{D}_g$ and $\mathcal{D}_{real}$. To measure the distance between distributions, we consider a general distance. Let $\mu, \nu$ be two distributions supported on $\mathbb{R}^{d_1}$:

$$d_\mathcal{F}(\mu, \nu) \triangleq \sup_{f \in \mathcal{F}} \mathbb{E}_{\mathbf{x} \sim \mu} [\phi(f(\mathbf{x}))] - \mathbb{E}_{\mathbf{x} \sim \nu} [\phi(f(\mathbf{x}))].$$

The objective function is equivalent to $\min_{g \in \mathcal{G}} d_\mathcal{F}(\mathcal{D}_{real}, \mathcal{D}_g)$. Without a loss of generality, we take $\phi(x) \triangleq x$ in our work. Therefore, our distribution distance can be revised as $d_\mathcal{F}(\mu, \nu) = \sup_{f \in \mathcal{F}} \mathbb{E}_{\mathbf{x} \sim \mu} [f(\mathbf{x})] - \mathbb{E}_{\mathbf{x} \sim \nu} [f(\mathbf{x})]$. Though the distribution distance is similar to Wasserstein Distance (Arjovsky et al., 2017), our discriminator function class $\mathcal{F}$ is not forced to be an 1-Lipschitz function class. According to Arora et al. (2017)'s work, if we constrain the range of $f$ to $[0, 1]$ and utilize the $\mathcal{F}$-distance (Arora et al., 2017), the representation can be reduced to the original GAN and WGAN by specific $\phi(\cdot)$. Thus, it can unify the JS divergence and the Wasserstein distance. Since our generalization theory also holds in these cases, we omit repetitive discussions of $\mathcal{F}$-distance.

**Rademacher Complexity**. The complexity or capacity of a network function class has a direct effect on the generalization properties of a network. Since a GAN model consists of two network structures, the complexities of $\mathcal{F}$ and $\mathcal{G}$ are the keys to further investigation of the generalization properties of GANs. The definition of the Rademacher complexity is given as follows. If we assume that $\mathcal{F}$ is a class of real value functions, and $\epsilon_i$ is the Rademacher Random Variable, then the empirical and expected Rademacher complexities are defined accordingly,

$$\hat{\mathfrak{R}}_S(\mathcal{F}) \triangleq \mathbb{E}_\epsilon \left[ \sup_{f \in \mathcal{N}} \frac{1}{n} \sum_{i=1}^n \epsilon_i f(\mathbf{z}_i) \right], \quad \mathfrak{R}_{n,\mathcal{D}}(\mathcal{F}) \triangleq \mathbb{E}_{S \sim \mathcal{D}^n} \left[ \hat{\mathfrak{R}}_S(\mathcal{F}) \right],$$

where $\epsilon_1, \ldots, \epsilon_n$ are independent Rademacher random variables, i.e., $\mathbb{P}(\epsilon_i = 1) = \mathbb{P}(\epsilon_i = -1) = 1/2$.

## 3 GENERALIZATION ERROR BOUND WITH A FIXED $g$

We first introduce the definition of generalization error for GANs. In supervised learning, generalization error refers to the gap between the training error and the test error. However, in the context of GANs, neither the training error nor the test error is well defined. This is because the discriminator $f$, which is the counterpart of the loss function, varies throughout the training process. In the following, we provide a reasonable measure for the counterparts of the training error and the test error for GANs using the distribution distance $d_{\mathcal{F}}(\cdot, \cdot)$. For a GAN model with the discriminator $f \in \mathcal{F}$ and the generator $g \in \mathcal{G}$, we define the training error as $d_{\mathcal{F}}(\hat{\mathcal{D}}_{real}, \mathcal{D}_g)$ and the real error as $d_{\mathcal{F}}(\mathcal{D}_{real}, \mathcal{D}_g)$, where $\hat{\mathcal{D}}_{real}$ is the empirical distribution over the sample set $\mathcal{S}$. The generalization error bound for GANs is defined as the difference between these two errors.

We compare our definition of the generalization error with that in Arora et al. (2017); Jiang et al. (2019); Zhang et al. (2017). Arora et al. (2017) defined the training error as $d_{\mathcal{F}}(\hat{\mathcal{D}}_{real}, \hat{\mathcal{D}}_g)$, which is related to the empirical distribution of $\mathcal{D}_g$. In other words, noise samples are regarded as a training set for GANs, while $\mathcal{D}_g$ is regarded as an unknown distribution. However, such a consideration is not commensurate with the experiment process. Instead of being reinput in every iteration like the training set $\mathcal{S}$, a new noise sample set of size $m$ is generated in every epoch. Thus, noise sample sets are not equivalent to the training set $\mathcal{S}$, since every noise sample sets is utilized only once.

The definition of the generalization bound in Jiang et al. (2019); Zhang et al. (2017) is slightly different from our intuition. In these works, $\bar{g}$ is defined as the generator, which is obtained by the training process, and $g^*$ is defined as the optimal generator for $\inf_{g \in \mathcal{G}} d_{\mathcal{F}}(\mathcal{D}_{real}, \mathcal{D}_g)$. Then, they define the generalization bound as $d_{\mathcal{F}}(\mathcal{D}_{real}, \mathcal{D}_{\bar{g}}) - d_{\mathcal{F}}(\mathcal{D}_{real}, \mathcal{D}_{g^*})$. However, in practise, exact values for $g^*$ and $\mathcal{D}_{real}$ are not accessible. Thus, the first term fails to represent the training error, and the second term fails to represent the testing error.

It is more reasonable to assume that $\mathcal{D}_g$ is a known distribution for a fixed $g$. The process of generating new noise samples in every epoch, is an empirical approximation of $\mathcal{D}_g$, rather than a simple "training noise" data collection process. From this perspective, our definition provides a more reasonable theoretical measure of the generalization error for GANs.

**Definition 3.1.** *For a GAN model with the discriminator $f \in \mathcal{F}$ and the generator $g \in \mathcal{G}$, the generalization error is defined as $|d_{\mathcal{F}}(\mathcal{D}_{real}, \mathcal{D}_g) - d_{\mathcal{F}}(\hat{\mathcal{D}}_{real}, \mathcal{D}_g)|$. We say $\epsilon$ is a generalization error bound if the following holds:*

$$\sup_{g \in \mathcal{G}} |d_{\mathcal{F}}(\mathcal{D}_{real}, \mathcal{D}_g) - d_{\mathcal{F}}(\hat{\mathcal{D}}_{real}, \mathcal{D}_g)| \leqslant \epsilon,$$

*where $\epsilon$ only relies on the parameter settings of $\mathcal{F}$ and $\mathcal{G}$.*

Intuitively, a low generalization error bound guarantees that the generalization error is low. Hence, the discriminator successfully discriminates between real data and unseen data. In these cases, the generator generates a distribution close to $\mathcal{D}_{real}$. An explicit generalization error bound provides us with guidance to design a GAN for designing a GAN to adequately fit real data. The following theorem provides an upper bound for the generalization error, with a fixed $g$.

**Theorem 3.2.** *For any fixed $g \in \mathcal{G}$, with a probability of at least $1 - \delta$ over the choice of samples $\mathcal{S}$:*

$$\left| d_{\mathcal{F}}(\mathcal{D}_{real}, \mathcal{D}_g) - d_{\mathcal{F}}(\hat{\mathcal{D}}_{real}, \mathcal{D}_g) \right| \leqslant 2\mathfrak{R}_{m, \mathcal{D}_{real}}(\mathcal{F}) + \sqrt{\frac{\log(1/\delta)}{m}}. \tag{1}$$

Though the result of Theorem 3.2 seems similar to the Theorem 3.1 in Zhang et al. (2017), our definition of generalization error is different. Theorem 3.2 shows that, for a fixed $g \in \mathcal{G}$, the bound for the generalization error mainly depends on the complexity of $\mathcal{F}$. Although the discriminator class should be complex enough to discriminate between $\mathcal{D}_{real}$ and $\mathcal{D}_g$, a grossly complicated discriminator class generates extra generalization errors. Theorem 3.2 also shows the relationship between $\mathfrak{R}_{m, \mathcal{D}_{real}}(\mathcal{F})$ and the generalization error bound. For some specific neural network function classes, such as the class of $\ell_{p,q}$ weight normalized neural networks, we can compute the Rademacher complexity to obtain an explicit upper bound for the generalization error. There are some previous works of Rademacher Complexity bounds, including Golowich et al. (2018); Neyshabur et al. (2015); Chen et al. (2019). We adopt Chen et al. (2019) to obtain a tighter and more general bound.

**Corollary 3.3.** *Assume* $\|\mathbf{x}\|_{p^*} \leqslant 1, \forall \mathbf{x} \in \mathcal{S}$. *For any fixed* $g \in \mathcal{G}$, *with a probability of at least* $1 - \delta$ *over the choice of samples* $\mathcal{S}$, *with* $\ell_{p,q}$ *weight normalization:*

$$\left| d_{\mathcal{F}}(\mathcal{D}_{real}, \mathcal{D}_g) - d_{\mathcal{F}}(\hat{\mathcal{D}}_{real}, \mathcal{D}_g) \right|$$

$$\leqslant 2\left( s_{k_1+1} \sqrt{\frac{(2k_1+4)\log 2}{m}} + \prod_{i=1}^{k_1+1} c_{\mathcal{F},i} \rho d_i^{[\frac{1}{p^*} - \frac{1}{q}]_+} d_0^{\frac{1}{p^*}} \sqrt{\frac{C(p)}{m}} \right) + \sqrt{\frac{\log(1/\delta)}{m}},$$

*where*

$$s_{k+1} \triangleq \sum_{i=1}^{k_1+1} \left( \prod_{l=i}^{k_1+1} c_{\mathcal{F},l} \rho d_l^{[\frac{1}{p^*} - \frac{1}{q}]_+} \right) + d_0^{\frac{1}{p^*}} \prod_{l=1}^{k_1+1} c_{\mathcal{F},l} \rho d_l^{[\frac{1}{p^*} - \frac{1}{q}]_+} \quad and$$

$$C(p) \triangleq \begin{cases} 2\log(2d_0) & p \in \{1\} \cup (2,\infty), \\ \min(p^*-1, 2\log(2d_0)) & p \in (1,2]. \end{cases}$$

By utilizing the big $O$ notation, Theorem 3.2 provides a bound of order $O(d_0^{\frac{1}{p^*}} \sqrt{k_1/m})$, where $1/p^* \leqslant 1$ usually holds. Notice that, some previous works (Bartlett et al., 2017; Jiang et al., 2019) provide bounds of order $O(\sqrt{d^3 k_1/m})$ and $O(d\sqrt{k_1/m})$, where $d = \max\{d_i\}_{i=1}^{k_1}$. Hence, our bound is tighter than previous works.

**Remark 3.4.** *A fair comparison between our bound and the bound in Arora et al. (2017) shows our bound is tighter. We denote $P_{\mathcal{F}}$ as the number of parameters of $f \in \mathcal{F}$, and $L_{\mathcal{F}}$ as the Lipschitz constant with respect to the parameters of $f \in \mathcal{F}$. We assess the two results under our definition of generalization bound in Section 3. According to Arora et al. (2017), if $m \geqslant 3P_{\mathcal{F}} \log(L_{\mathcal{F}} P_{\mathcal{F}}/\epsilon)/\epsilon^2$, we have a probability of at least $1 - \exp(-P_{\mathcal{F}})$ over the choice of $\mathcal{S}$, $|d_{\mathcal{F}}(\mathcal{D}_{real}, \mathcal{D}_g) - d_{\mathcal{F}}(\hat{\mathcal{D}}_{real}, \mathcal{D}_g)| \leqslant \epsilon/2$. This holds because the formula (6) in the Theorem B.2 of Arora et al. (2017) no longer contributes to the generalization bound. We convert our result into a similar fashion by utilizing $1/m \leqslant \epsilon^2/(3P_{\mathcal{F}} \log(L_{\mathcal{F}} P_{\mathcal{F}}/\epsilon))$ and $\delta = \exp(-P_{\mathcal{F}})$. We obtain*

$$|d_{\mathcal{F}}(\mathcal{D}_{real}, \mathcal{D}_g) - d_{\mathcal{F}}(\hat{\mathcal{D}}_{real}, \mathcal{D}_g)|$$

$$\leqslant \left( 2(s_{k_1+1} \sqrt{(2k_1+4)\log 2} + \prod_{i=1}^{k_1+1} c_{\mathcal{F},i} \rho d_i^{[\frac{1}{p^*} - \frac{1}{q}]_+} d_0^{\frac{1}{p^*}} \sqrt{C(p)}) + \sqrt{P_{\mathcal{F}}} \right) \cdot \frac{\epsilon}{\sqrt{3P_{\mathcal{F}} \log(L_{\mathcal{F}} P_{\mathcal{F}}/\epsilon)}}.$$

*Since $\{c_{\mathcal{F},i}\}_{i=1}^{k_1}$ can be constrained to small values by applying weight normalization, we can force the following holds:*

$$2(s_{k_1+1} \sqrt{(2k_1+4)\log 2} + \prod_{i=1}^{k_1+1} c_{\mathcal{F},i} \rho d_i^{[\frac{1}{p^*} - \frac{1}{q}]_+} d_0^{\frac{1}{p^*}} \sqrt{C(p)})$$

$$\leqslant \sqrt{P_{\mathcal{F}}} \cdot \left( \frac{1}{2} \sqrt{3\log(\sum_{i=1}^{k_1+1} \rho^{k_1+1-i} \prod_{j=i}^{l} d_l^{[\frac{1}{p^*} - \frac{1}{q}]_+} c_{\mathcal{F},j} P_{\mathcal{F}}/\epsilon)} - 1 \right).$$

$\log(L_{\mathcal{F}} P_{\mathcal{F}}/\epsilon) \approx \log(P_{\mathcal{F}} \sqrt{m})$ *and $P_{\mathcal{F}}$ are large numbers, which only depend on the structure of $\mathcal{F}$. Hence, there exits $\{c_{\mathcal{F},i}\}_{i=1}^{k_1}$ that is small enough to satisfy the inequality above. In these cases, our bound is tighter than Arora et al. (2017)'s bound. In fact, the weight normalization contracts the range of network parameters, so the complexity of $\mathcal{F}$ is reduced, thereby leading to a tighter generalization error bound.*

Inspired by the probabilistic inequality in corollary 3.3, we formulate a hypothesis testing process to judge whether a generator produces data with the same distribution as that of the real data. The appendix contains the theory and experiments related to this novel hypothesis test on a toy dataset.

If we adopt ReLU (a homogeneous function) as an active function and apply the $\ell_{p,q}$ weight normalization, the bound for the generalization error can be further reduced to become width-independent.

**Corollary 3.5.** *Under the same settings as Corollary 3.3, if $1/p + 1/q \geqslant 1$, we adopt the ReLU function as active function. Then, for any fixed $g \in \mathcal{G}$, with a probability of at least $1 - \delta$ over the choice of samples $\mathcal{S}$:*

$$\left| d_{\mathcal{F}}(\mathcal{D}_{real}, \mathcal{D}_g) - d_{\mathcal{F}}(\hat{\mathcal{D}}_{real}, \mathcal{D}_g) \right|$$

$$\leqslant 2\left( (1 + d_0^{\frac{1}{p^*}}) \prod_{l=1}^{k_1+1} c_{\mathcal{F},l} \sqrt{\frac{(2k_1+4)\log 2}{m}} + \prod_{i=1}^{k_1+1} c_{\mathcal{F},i} d_0^{\frac{1}{p^*}} \sqrt{\frac{C(p)}{m}} \right) + \sqrt{\frac{\log(1/\delta)}{m}}.$$

We can easily extend our results to cases with spectral weight normalization (Jiang et al., 2019).

**Corollary 3.6.** *Let $d = \max\{d_i\}_{i=1}^{k_1+1}$. Under the same settings as Corollary 3.3, for any fixed $g \in \mathcal{G}$, with a probability of at least $1 - \delta$ over the choice of samples $\mathcal{S}$ with spectral weight normalization:*

$$\left| d_{\mathcal{F}}(\mathcal{D}_{real}, \mathcal{D}_g) - d_{\mathcal{F}}(\hat{\mathcal{D}}_{real}, \mathcal{D}_g) \right|$$

$$\leqslant \frac{24(\prod_{i=1}^{k_1+1} B_{\mathcal{F},i})d\sqrt{k_1 \log(2\sqrt{dm}k_1 \prod_{i=1}^{k_1+1} B_{\mathcal{F},i})}}{\sqrt{m}} + \frac{8}{\sqrt{m}} + \sqrt{\frac{\log(1/\delta)}{m}}.$$

Corollary 3.6 shows the advantage of applying spectral weight normalization. We constrain $\prod_{i=1}^{k_1+1} B_{\mathcal{F},i}$ such that it is small enough to force the first term to be small. If we set $\prod_{i=1}^{k_1+1} B_{\mathcal{F},i} = 1$, the conclusion is reduced to $O\{\sqrt{d^2 k_1/m}\}$. Experiments in Miyato et al. (2018) also show that, spectral weight normalization render the discriminator more powerful for distinguishing between generated data and real data. Hence, we can suffer less from model collapse. Since $d$ depends on the largest width of the network, the bound in spectral normalization is not actually width-independent. For this reason, we prefer to utilize the $\ell_{p,q}$ weight normalization to obtain a width-independent bound.

## 4 UNIFORM GENERALIZATION ERROR BOUND

We have established the generalization error upper bound with a fixed generator $g$, that is, $g$ is independent of the training process and the choice of sample set $\mathcal{S}$. However, such a bound is not a uniform bound for $\forall g \in \mathcal{G}$. For any fixed $g$, let $\mathcal{S}(g)$ denote the set of samples where the inequality (1) in Theorem 3.2 holds, i.e., $\mathcal{S}(g) \triangleq \{\mathcal{S} \mid \mathcal{S} \overset{i.i.d}{\sim} \mathcal{D}_{real}^m, \text{ bound (1) holds with } \mathcal{S}\}$. We emphasize the dependence of $\mathcal{S}(g)$ on $g$ because this set varies as $g$ changes. In other words, different $\mathcal{S}(g)$ values lead to different probability levels, at which the bound holds. Hence, the probability that (1) holds with $\mathcal{S} \in \cap_{g \in \mathcal{G}} \mathcal{S}(g)$ is not guaranteed to be greater than $1 - \delta$. An upper bound for the generalization error with a fixed $g$ is tantamount to the generalization error bound for neural networks in supervised learning. To further understand GANs, it is necessary for us to establish a uniform bound for generalization error with varying $g$. The following theorem provides a generalization bound for GANs.

**Theorem 4.1.** *With a probability of at least $1 - 2|\mathcal{X}| \cdot \exp(-m\epsilon^2/4)$ over the choice of samples $\mathcal{S}$:*

$$\sup_{g \in \mathcal{G}} |d_{\mathcal{F}}(\mathcal{D}_{real}, \mathcal{D}_g) - d_{\mathcal{F}}(\hat{\mathcal{D}}_{real}, \mathcal{D}_g)| \leqslant 2\mathfrak{R}_{m,\mathcal{D}_{real}}(\mathcal{F}) + \epsilon,$$

*where $\mathcal{X}$ is a $\frac{\epsilon}{2LL_{\mathcal{G}}}$-net of $\mathcal{V}$, and $\mathcal{V}$ is the parameter space of $\mathcal{G}$.*

Theorem 4.1 depicts the error that is contributed by $\mathcal{G}$. Note that, if $\mathcal{G}$ is complicated, then $|\mathcal{X}|$ can be a large number. In other words, if we reduce the complexity of $\mathcal{G}$ by applying weight normalization, the generalization error bound will consequentially decrease. In fact, weight normalization is an approach for reducing the complexity of a function class and provides generalization error bound control. For the $\ell_{p,q}$ weight normalization, the next theorem provides an explicit expression of the generalization error bound.

**Corollary 4.2.** *With a probability of at least $1 - \delta$ over the choice of samples $\mathcal{S}$:*

$$\sup_{g \in \mathcal{G}} |d_{\mathcal{F}}(\mathcal{D}_{real}, \mathcal{D}_g) - d_{\mathcal{F}}(\hat{\mathcal{D}}_{real}, \mathcal{D}_g)|$$

$$\leqslant 2\left( s_{k_1+1}\sqrt{\frac{(2k_1+4)\log 2}{m}} + \prod_{i=1}^{k_1+1} c_{\mathcal{F},i}\rho d_i^{[\frac{1}{p^*}-\frac{1}{q}]_+} d_0^{\frac{1}{p^*}}\sqrt{\frac{C(p)}{m}} \right) + c_\delta\sqrt{\frac{2P_{\mathcal{G}}}{m}\log(6k_2 LL_{\mathcal{G}})},$$

*where $c_\delta$ satisfies*

$$\log(1/\delta) \leqslant (\frac{c_\delta^2}{2} - 1)P_{\mathcal{G}}\log(6k_2 LL_{\mathcal{G}}) + P_{\mathcal{G}}\log(c_\delta\sqrt{\frac{2P_{\mathcal{G}}}{m}\log(6k_2 LL_{\mathcal{G}})},$$

*$s_{k_1+1}$ and $C(p)$ are given in theorem 3.3, and $P_{\mathcal{G}}$ denotes the number of parameters in $g \in \mathcal{G}$.*

Notice that with the $\ell_{p,q}$ weight normalization, we obtain an upper bound for $L, L_{\mathcal{G}}$, in a $\ell_{p,q}$ norm version: $L \leqslant \prod_{i=1}^{k_1+1} c_{\mathcal{F},i} \rho d_i^{[\frac{1}{p^*}-\frac{1}{q}]_+}, L_{\mathcal{G}} \leqslant \sum_{i=1}^{k_2+1} \rho^{k_2+1-i} \prod_{j=i}^{k_2+1} l_j^{[\frac{1}{p^*}-\frac{1}{q}]_+} c_{\mathcal{G},j}$. Hence, for an arbitrary weight normalized GAN, we calculate the generalization bound with $P_{\mathcal{G}}, c_{\mathcal{F},i}$ and $c_{\mathcal{G},j}$. For every $\delta$, we calculate the right hand side of the constraint and pick a feasible and small $c_\delta$. According to theorem 4.2, the choice of $g$ contributes to the second term. Since the bounds $\{c_{\mathcal{F},i}\}_{i=1}^{k_1}$ for weight normalization can be artificially constrained to a small range, the first two terms can be constrained to a small value. Since $P_{\mathcal{G}}$ is determined by the network structure of $g \in \mathcal{G}$ and is usually much larger than $k_2^2$, the third term is dominate. Similarly, we obtain a width-independent generalization bound by adopting ReLU functions and applying specific $\ell_{p,q}$ weight normalization, with $1/p + 1/q \geqslant 1$.

Our result can be extended to cases with spectral weight normalization.

**Corollary 4.3.** *Assume $\|\mathbf{x}\|_2 \leqslant 1$ for $\mathbf{x} \in \mathcal{S}$. With a probability of at least $1 - \delta$ over the choice of samples $\mathcal{S}$ with spectral weight normalization:*

$$\sup_{g \in \mathcal{G}} |d_{\mathcal{F}}(\mathcal{D}_{real}, \mathcal{D}_g) - d_{\mathcal{F}}(\hat{\mathcal{D}}_{real}, \mathcal{D}_g)| \leqslant 24 \prod_{i=1}^{k_1+1} B_{\mathcal{F},i} d \sqrt{\frac{k_1}{m} \log(2\sqrt{dm}k_1 \prod_{i=1}^{k_1+1} B_{\mathcal{F},i})}$$

$$+ \frac{8}{\sqrt{m}} + c_\delta \sqrt{\frac{P_{\mathcal{G}}}{m} \log(6k_2 \prod_{i=1}^{k_1+1} B_{\mathcal{F},i} \prod_{j=i}^{k_2+1} B_{\mathcal{G},j})},$$

*where $c_\delta$ satisfies*

$$\log(1/\delta) \leqslant (\frac{c_\delta^2}{2} - 1) P_{\mathcal{G}} \log(6k_2 \prod_{i=1}^{k_1+1} B_{\mathcal{F},i} \prod_{j=i}^{k_2+1} B_{\mathcal{G},j})$$

$$+ P_{\mathcal{G}} \log(c_\delta \sqrt{\frac{2P_{\mathcal{G}}}{m} \log(6k_2 \prod_{i=1}^{k_1+1} B_{\mathcal{F},i} \prod_{j=i}^{k_2+1} B_{\mathcal{G},j})}).$$

## 5 NUMERICAL EXPERIMENTS.

In this section, we illustrate some numerical experiments to verify that our generalization error bound is consistent with numerical studies. We train Wasserstein generative adversarial networks (WGANs) to learn a three-Gaussian Mixture distribution. The structure of the discriminator is a three-layer $(2 \times 50$ FC)-ReLU-$(50 \times 50$ FC)-ReLU-$(50 \times 50$ FC)-ReLU-$(50 \times 1$ FC) network, where FC denotes a fully connected layer. The generator is a three-layer $(2 \times d$ FC)-ReLU-$(d \times 50$ FC)-ReLU-$(50 \times 50$ FC)-ReLU-$(50 \times 2$ FC) network, where $d$ takes value in $\{50, 70, 90, 110, 130\}$. After the training process, we calculate the generalization error and generalization error bound. The details of the experimental settings are included in the appendix.

In order to compute the generalization error with an empirical approach, we generate two data sets, $\mathcal{S}^{train}$ and $\mathcal{S}^{test}$, while the sample size of $\mathcal{S}^{test}$ is much larger than that of $\mathcal{S}^{train}$. We regard the empirical distribution over $\mathcal{S}^{test}$ as an approximation of $\mathcal{D}_{real}$. The noise is generated from $\mathbf{z} \sim N_2(\mathbf{0}, \mathbf{I})$, in the training process.

We compare the generalization error and generalization bound of the WGAN model. We adopt an empirical approach to calculate $d_{\mathcal{F}}(\mathcal{D}_{real}, \mathcal{D}_g)$ and $d_{\mathcal{F}}(\hat{\mathcal{D}}_{real}, \mathcal{D}_g)$. Since theorem 4.2 provides a computable generalization bound with $\ell_{2,2}$ weight normalization, for a trained WGAN, we can calculate the generalization error and generalization bound. For each $d \in \{50, 70, 90, 110, 130\}$, we repeat the following process 50 times to compute the generalization error and the average generalization error. At the same time, we calculate the generalization error bound for each $d$ and the average value. Figure 1 displays two visualizations of the simulated results. The left panel shows a visibly good generator and the right panel shows a visibly bad generator.

According to the previous discussions, generalization error is defined as $|d_{\mathcal{F}}(\mathcal{D}_{real}, \mathcal{D}_g) - d_{\mathcal{F}}(\hat{\mathcal{D}}_{real}, \mathcal{D}_g)|$. To compute $d_{\mathcal{F}}(\widehat{\mathcal{D}_{real}}, \mathcal{D}_g), d_{\mathcal{F}}(\widehat{\hat{\mathcal{D}}_{real}}, \mathcal{D}_g)$, we approximate the distribution distance $d_{\mathcal{F}}(\mathcal{D}_{real}, \mathcal{D}_g), d_{\mathcal{F}}(\hat{\mathcal{D}}_{real}, \mathcal{D}_g)$ by substituting $\mathcal{D}_{real}, \hat{\mathcal{D}}_{real}$ with the uniform distribution over

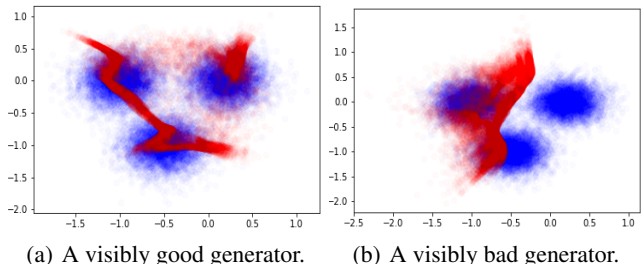

(a) A visibly good generator.     (b) A visibly bad generator.

Figure 1: The blue point cloud represents $\mathcal{S}^{train}$, a sample from a Gaussian Mixture distribution. The red points are $\{g(\mathbf{z}_i)\}_{i=1}^{m_{train}}$, while $\{\mathbf{z}_i\}_{i=1}^{m_{train}} \overset{i.i.d}{\sim} N_2(\mathbf{0}, \mathbf{I})$.

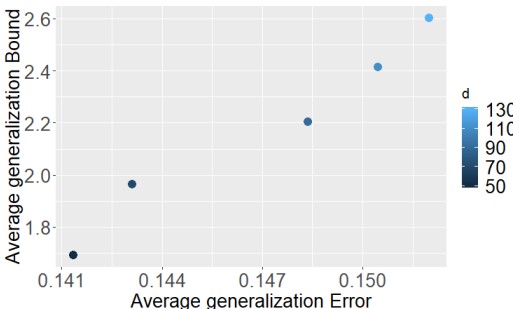

Figure 2: Each dot on the graph represents the average generalization error and the generalization bound of WGAN with $d \in \{50, 70, 90, 110, 130\}$.

$\mathcal{S}_{test}, \mathcal{S}_{train}$ (denoted as $\hat{\mathcal{D}}_{test}, \hat{\mathcal{D}}_{train}$), respectively. The detailed process is included in the appendix.

According to figure 2, there is a positive correlation between the generalization error and generalization bound. As the number of parameters in the generator increases, the generalization bound increases. In other words, the experiment verifies that, our bound provides generalization error control. A low generalization bound guarantees that the generalization error is low, whereas a generator with a large number of parameters introduces more error.

In fact, by applying $\ell_{p,q}$ weight normalization to $f, g$ with small $c_{\mathcal{F},i}, c_{\mathcal{G},j}$, we control the generalization bound so that it remains a small value. The number of parameters in the generator should not be extremely large. So far, the experiment and our theory have explained how $\ell_{p,q}$ weight normalization and the parameter settings of $f, g$ affect the generalization of GANs. Our generalization bound provides explicit guidance on parameters designing to train GANs with small generalization error. Thus, we can obtain robust generators.

## 6   CONCLUSION

In this paper, we establish the generalization theory for GANs and provide a more reasonable definition of generalization error. We first establish a general bound for generalization error, with a fixed generator. To further understand GANs, we establish the generalization error bound, which uniformly holds over any choice of generators. Our numerical experiments on Gaussian Mixture models verify that, our theory is consistent with the numerical studies. In the Appendix, we also formulate a novel hypothesis testing procedure to judge whether the generated distribution equals the distrbution of observed data. Notice that, in these high dimension cases, the ordinary statistical approaches do not work well. Our hypothesis test is capable of discriminating between good and bad generators. One interesting future research topic is to develop generalization error bounds for autoencoder GANs with an additional encoder network.

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

## A    PROOF DETAILS FOR MAIN RESULTS

PROOF OF THEOREM 3.2

*Proof.*

$$\left| \sup_{f \in \mathcal{F}} \left[ \mathop{\mathbb{E}}_{x \sim \mathcal{D}} f(x) - \mathop{\mathbb{E}}_{z \sim \mathcal{N}} f \circ g(z) \right] - \sup_{f \in \mathcal{F}} \left[ \mathop{\mathbb{E}}_{x \sim \hat{\mathcal{D}}} f(x) - \mathop{\mathbb{E}}_{z \sim \mathcal{N}} f \circ g(z) \right] \right|$$

$$\leqslant \sup_{f \in \mathcal{F}} \left| \left[ \mathop{\mathbb{E}}_{x \sim \mathcal{D}} f(x) - \mathop{\mathbb{E}}_{z \sim \mathcal{N}} f \circ g(z) \right] - \left[ \mathop{\mathbb{E}}_{x \sim \hat{\mathcal{D}}} f(x) - \mathop{\mathbb{E}}_{z \sim \mathcal{N}} f \circ g(z) \right] \right|$$

$$= \sup_{f \in \mathcal{F}} \left| \mathop{\mathbb{E}}_{x \sim \mathcal{D}} f(x) - \mathop{\mathbb{E}}_{x \sim \hat{\mathcal{D}}} f(x) \right|$$

Using the well-known estimation method for the generalization bound in supervised learning Mohri et al. (2018), with a probability of at least $1 - \delta$ over the choice of samples, we have

$$\sup_{f \in \mathcal{F}} \left| \mathop{\mathbb{E}}_{x \sim \mathcal{D}} f(x) - \mathop{\mathbb{E}}_{x \sim \hat{\mathcal{D}}} f(x) \right| \leqslant 2 \mathfrak{R}_{m, \mathcal{D}_{real}}(\mathcal{F}) + \sqrt{\frac{\log(1/\delta)}{m}}$$

PROOF OF COROLLARY 3.3

We adopt Chen et al. (2019) to obtain a tighter and more general bound of the Rademacher Complexity:

$$\mathfrak{R}_m(\mathcal{F})$$
$$\leqslant s_{k_1+1} \sqrt{\frac{(2k_1 + 4)\log 2}{m}} + \prod_{i=1}^{k_1+1} c_{\mathcal{F},i} \rho d_i^{[\frac{1}{p^*} - \frac{1}{q}]_+} d_0^{\frac{1}{p^*}} \sqrt{\frac{C(p)}{m}}$$

Where

$$s_{k_1+1} \triangleq \sum_{i=1}^{k_1+1} \left( \prod_{l=i}^{k_1+1} c_{\mathcal{F},l} \rho d_l^{[\frac{1}{p^*} - \frac{1}{q}]_+} \right) + d_0^{\frac{1}{p^*}} \prod_{l=1}^{k_1+1} c_{\mathcal{F},l} \rho d_l^{[\frac{1}{p^*} - \frac{1}{q}]_+}$$

$$C(p) \triangleq \begin{cases} 2\log(2d_0) & p = 1 \,, \\ \min(p^* - 1, 2\log(2d_0)) & p > 1 \,. \end{cases}$$

$\square$

PROOF OF THEOREM 4.1

**Lemma A.1.** $\forall \|\mathbf{x}\|_{p^*} \leqslant 1, f \in \mathcal{F}$, *we have* $f(\mathbf{x}) \leqslant c_{\mathcal{F},k_1+1} \sum_{i=1}^{k_1} \rho^{k_1+1-i} \prod_{j=i}^{k_1} d_j^{[\frac{1}{p^*} - \frac{1}{q}]_+} c_{\mathcal{F},j}$. *Where $\rho$ is the Lipschitz constant of the active function.*

*Proof.* Define a series of variables $\{Z_0, Z_1, ..., Z_{k_1}\}$ as

$$Z_0 = \|\mathbf{x}\|_{p^*}$$

and

$$Z_{l+1} = \|f_{l+1}(\mathbf{x})\|_{p^*}, l = 0, 1, ..., k_1,$$

where $f_l$ is the combination of the first $l$ layers. Then, we prove by induction that for $l = 1, ..., k_1$,

$$Z_l \leqslant \sum_{i=1}^{l} \rho^{l+1-i} \prod_{j=i}^{l} d_j^{[\frac{1}{p^*} - \frac{1}{q}]_+} c_{\mathcal{F},j}.$$

When $l = 0$, we have $Z_0 = \|\mathbf{x}\|_{p^*} \leqslant 1$. For $l = 1, 2, ..., k_1$,

$$\begin{aligned}
Z_l =& \|f_l(\mathbf{x})\|_{p^*} \\
=& \|\sigma(\mathbf{W}_{f,l}(\mathbf{x}) + \mathbf{b}_{f,l})\|_{p^*} \\
\leqslant& \rho d_l^{[\frac{1}{p^*} - \frac{1}{q}]_+} (\|\mathbf{W}_{f,l}\|_{p,q}\|f_{l-1}(\mathbf{x})\|_{p^*} + \|\mathbf{b}_{f,l}\|_{p^*}) \\
\leqslant& \rho d_l^{[\frac{1}{p^*} - \frac{1}{q}]_+} (c_{\mathcal{F},l} Z_{l-1} + c_{\mathcal{F},l}) \\
\leqslant& \sum_{i=1}^{l} \rho^{l+1-i} \prod_{j=i}^{l} d_j^{[\frac{1}{p^*} - \frac{1}{q}]_+} c_{\mathcal{F},j}
\end{aligned}$$

Finally, we have

$$\begin{aligned}
& f(\mathbf{x}) \\
=& \|\mathbf{W}_{f,k_1+1} f_{k_1}(\mathbf{x})\|_{p^*} \\
\leqslant& \|\mathbf{W}_{f,k_1+1}\|_{p^*} \|f_{k_1}(\mathbf{x})\|_{p^*} \\
\leqslant& c_{\mathcal{F},k_1+1} \sum_{i=1}^{k_1} \rho^{k_1+1-i} \prod_{j=i}^{k_1} d_j^{[\frac{1}{p^*} - \frac{1}{q}]_+} c_{\mathcal{F},j}
\end{aligned}$$

$\square$

**Lemma A.2.** *For generator class $\mathcal{G}$, which is parameterized as the previous discussions, there is a $\epsilon/L_\mathcal{G}$-net $\mathcal{X}$ for parameter space $\mathcal{V}$, with respect to the $\ell_{p,q}$ parameter norm, such that*

$$\log |\mathcal{X}| \leqslant P_\mathcal{G} \log(3k_2 L_\mathcal{G}/\epsilon),$$

*where $L_\mathcal{G} = c_{\mathcal{G},k_2+1} \sum_{i=1}^{k_2} \rho^{k_2+1-i} \prod_{j=i}^{k_2} l_j^{[\frac{1}{p^*} - \frac{1}{q}]_+} c_{\mathcal{G},j}$.*

*Proof.* For two generators $g_\mathbf{v}$, $g_{\mathbf{v}'} \in \mathcal{G}$, $\mathbf{v} = (\mathbf{M}_{g,k_2+1}, \ldots, \mathbf{M}_{g,1}), \mathbf{v}' = (\mathbf{M}'_{g,k_2+1}, \ldots, \mathbf{M}'_{g,1}) \in \mathcal{V}$, we have

$$\begin{aligned}
& |g_\mathbf{v}(\mathbf{x}) - g_{\mathbf{v}'}(\mathbf{x})| \\
\leqslant& |g_{\{\mathbf{M}_{g,k_2+1}, \ldots, \mathbf{M}_{g,1}\}} - g_{\{\mathbf{M}'_{g,k_2+1}, \mathbf{M}_{g,k_2} \ldots, \mathbf{M}_{g,1}\}}| + \cdots \\
& + |g_{\{\mathbf{M}'_{g,k_2+1}, \ldots, \mathbf{M}'_{g,2}, \mathbf{M}_{g,1}\}} - g_{\{\mathbf{M}'_{g,k_2+1}, \ldots, \mathbf{M}'_{g,1}\}}| \\
\leqslant& L_\mathcal{G} \frac{\|(\Delta\mathbf{M}_{g,k_2+1})^\top\|_{p,q}}{c_{\mathcal{G},k_2+1}} + \cdots + L_\mathcal{G} \frac{\|(\Delta\mathbf{M}_{g,1})^\top\|_{p,q}}{c_{\mathcal{G},1}} \\
=& L_\mathcal{G} \sum_{i=1}^{k_2+1} \frac{\|(\Delta\mathbf{b}_{g,i}, \Delta\mathbf{W}_{g,i}^\top)^\top\|_{p,q}}{c_{\mathcal{G},i}},
\end{aligned}$$

Notice that, we derive the second inequality by applying lemma A.1 on $g$.
Then, we focus on the $\frac{c_{\mathcal{G},i}}{k_2 L_\mathcal{G}} \epsilon$-nets of set $\{\mathbf{x} \in \mathbb{R}^{l_{i-1} \times l_i} \mid \|\mathbf{x}\|_{p,q} \leqslant c_{\mathcal{G},i}\}$. According to lemma 1.3 of Berthet, there exists a $\frac{c_{\mathcal{G},i}}{k_2 L_\mathcal{G}} \epsilon$-net $\mathcal{X}_i$ that satisfyies $\log |\mathcal{X}_i| \leqslant l_{i-1}(l_i+1) \log(3k_2 L_\mathcal{G}/\epsilon)$. We construct a $\epsilon/L_\mathcal{G}$-net $\mathcal{X}$ of $\mathcal{V}$, by taking $\mathcal{X} \triangleq \{\mathbf{v} = (\mathbf{M}_{g,k_2+1}, \ldots, \mathbf{M}_{g,1}) \in \mathcal{V} \mid \mathbf{M}_{g,i} \in \mathcal{X}_i, i = 1, ..., k_2+1\}$. Hence, we have $\log |\mathcal{X}| \leqslant P_\mathcal{G} \log(3k_2 L_\mathcal{G}/\epsilon)$. $\square$

**Lemma A.3.** *For any generator $g \in \mathcal{G}$,*

$$\left| d_\mathcal{F}(\mathcal{D}_{real}, \mathcal{D}_g) - \mathbb{E}_\mathcal{S}[d_\mathcal{F}(\hat{\mathcal{D}}_{real}, \mathcal{D}_g)] \right| \leqslant 2\mathfrak{R}_{m,\mathcal{D}_{real}}(\mathcal{F})$$

*Proof.*

$$
\left| d_{\mathcal{F}}(\mathcal{D}_{real}, \mathcal{D}_g) - \mathbb{E}_{\mathcal{S}}[d_{\mathcal{F}}(\hat{\mathcal{D}}_{real}, \mathcal{D}_g)] \right|
$$

$$
= \left| \sup_{f \in \mathcal{F}} \left[ \mathop{\mathbb{E}}_{\mathbf{x} \sim \mathcal{D}_{real}} f(\mathbf{x}) - \mathop{\mathbb{E}}_{\mathbf{z} \sim \mathcal{N}} f \circ g(\mathbf{z}) \right] - \mathbb{E}_{\mathcal{S}} \sup_{f \in \mathcal{F}} \left[ \mathop{\mathbb{E}}_{\mathbf{x} \sim \hat{\mathcal{D}}_{real}} f(\mathbf{x}) - \mathop{\mathbb{E}}_{\mathbf{z} \sim \mathcal{N}} f \circ g(\mathbf{z}) \right] \right|
$$

$$
\leqslant \mathbb{E}_{\mathcal{S}} \left| \sup_{f \in \mathcal{F}} \left[ \mathop{\mathbb{E}}_{\mathbf{x} \sim \mathcal{D}_{real}} f(\mathbf{x}) - \mathop{\mathbb{E}}_{\mathbf{z} \sim \mathcal{N}} f \circ g(\mathbf{z}) \right] - \sup_{f \in \mathcal{F}} \left[ \mathop{\mathbb{E}}_{\mathbf{x} \sim \hat{\mathcal{D}}_{real}} f(\mathbf{x}) - \mathop{\mathbb{E}}_{\mathbf{z} \sim \mathcal{N}} f \circ g(\mathbf{z}) \right] \right|
$$

$$
\leqslant \mathbb{E}_{\mathcal{S}} \sup_{f \in \mathcal{F}} \left| \left[ \mathop{\mathbb{E}}_{\mathbf{x} \sim \mathcal{D}_{real}} f(\mathbf{x}) - \mathop{\mathbb{E}}_{\mathbf{z} \sim \mathcal{N}} f \circ g(\mathbf{z}) \right] - \left[ \mathop{\mathbb{E}}_{\mathbf{x} \sim \hat{\mathcal{D}}_{real}} f(\mathbf{x}) - \mathop{\mathbb{E}}_{\mathbf{z} \sim \mathcal{N}} f \circ g(\mathbf{z}) \right] \right|
$$

$$
\leqslant 2 \mathfrak{R}_{m, \mathcal{D}_{real}}(\mathcal{F}),
$$

where we denote the noise distribution $\mathcal{N}_{l_0}(\mathbf{0}, \mathbf{I})$ as $\mathcal{N}$ for short. □

**Lemma A.4.** *With $\mathcal{F}, \mathcal{G}$ fixed, the following holds with a probability of at least $1 - 2\delta$ over the choice of samples $\mathcal{S}$:*

$$
\sup_{g \in \mathcal{G}} \left| \mathbb{E}_{\mathcal{S}} \sup_{f \in \mathcal{F}} \left[ \mathop{\mathbb{E}}_{\mathbf{x} \sim \hat{\mathcal{D}}_{real}} f(\mathbf{x}) - \mathop{\mathbb{E}}_{\mathbf{z} \sim \mathcal{N}} f \circ g(\mathbf{z}) \right] - \sup_{f \in \mathcal{F}} \left[ \mathop{\mathbb{E}}_{\mathbf{x} \sim \hat{\mathcal{D}}_{real}} f(\mathbf{x}) - \mathop{\mathbb{E}}_{\mathbf{z} \sim \mathcal{N}} f \circ g(\mathbf{z}) \right] \right|
$$

$$
\leqslant \frac{c_\delta \sqrt{2 P_{\mathcal{G}} \log(3 k_2 L L_{\mathcal{G}})}}{\sqrt{m}},
$$

*where $c_\delta$ satisfies*

$$
\log(1/\delta) \leqslant (\frac{c_\delta^2}{2} - 1) P_{\mathcal{G}} \log(3 k_2 L L_{\mathcal{G}}) + P_{\mathcal{G}} \log(c_\delta \sqrt{2 P_{\mathcal{G}} \log(3 k_2 L L_{\mathcal{G}})/m}).
$$

*Proof.* Let $\mathcal{S}_1$ and $\mathcal{S}_1'$ be two sample sets from real data, with $\#\mathcal{S}_1 = \#\mathcal{S}_1' = m$. They differ by exactly one element, which is denoted as $\mathbf{x}_i \in \mathcal{S}_1$ and $\mathbf{x}_i' \in \mathcal{S}_1'$, respectively. For a fixed $g$, we have

$$
\sup_{f \in \mathcal{F}} \left[ \mathop{\mathbb{E}}_{\mathbf{x} \sim \hat{\mathcal{D}}} f(\mathbf{x}) - \mathop{\mathbb{E}}_{\mathbf{z} \sim \mathcal{N}} f \circ g(\mathbf{z}) \right] - \sup_{f \in \mathcal{F}} \left[ \mathop{\mathbb{E}}_{\mathbf{x} \sim \hat{\mathcal{D}}'} f(x) - \mathop{\mathbb{E}}_{\mathbf{z} \sim \mathcal{N}} f \circ g(\mathbf{z}) \right]
$$

$$
\leqslant \sup_{f \in \mathcal{F}} \left| \left[ \mathop{\mathbb{E}}_{\mathbf{x} \sim \hat{\mathcal{D}}} f(\mathbf{x}) - \mathop{\mathbb{E}}_{\mathbf{z} \sim \mathcal{N}} f \circ g(\mathbf{z}) \right] - \left[ \mathop{\mathbb{E}}_{\mathbf{x} \sim \hat{\mathcal{D}}'} f(\mathbf{x}) - \mathop{\mathbb{E}}_{\mathbf{z} \sim \mathcal{N}} f \circ g(\mathbf{z}) \right] \right|
$$

$$
= \sup_{f \in \mathcal{F}} \left| \left[ \mathop{\mathbb{E}}_{\mathbf{x} \sim \hat{\mathcal{D}}} f(\mathbf{x}) - \mathop{\mathbb{E}}_{\mathbf{x} \sim \hat{\mathcal{D}}'} f(x) \right] \right|
$$

$$
= \sup_{f \in \mathcal{F}} \left| \frac{1}{m} \left( f(\mathbf{x}_i) - f(\mathbf{x}_i') \right) \right| \leqslant \frac{2}{m},
$$

where $\mathcal{D}, \mathcal{D}'$ stands for the uniform distribution over $\mathcal{S}_1, \mathcal{S}_1'$. According to McDiarmid's inequality, it holds that:

$$
\mathbb{P}_{\mathcal{S}} \left[ \left| \mathbb{E}_{\mathcal{S}} \sup_{f \in \mathcal{F}} \left[ \mathop{\mathbb{E}}_{\mathbf{x} \sim \hat{\mathcal{D}}} f(\mathbf{x}) - \mathop{\mathbb{E}}_{\mathbf{z} \sim \mathcal{N}} f \circ g(\mathbf{z}) \right] - \sup_{f \in \mathcal{F}} \left[ \mathop{\mathbb{E}}_{\mathbf{x} \sim \hat{\mathcal{D}}} f(\mathbf{x}) - \mathop{\mathbb{E}}_{\mathbf{z} \sim \mathcal{N}} f \circ g(\mathbf{z}) \right] \right| \geqslant \frac{\epsilon}{2} \right]
$$

$$
\leqslant 2 \exp(-\frac{\epsilon^2/2}{2/m}),
$$

where $\hat{\mathcal{D}}$ stands for the uniform distribution over $\mathcal{S}$. By following Lemma A.2, we let $\mathcal{X}$ be a $\frac{\epsilon}{2 L L_{\mathcal{G}}}$-net of $\mathcal{V}$ that satisfies $\log |\mathcal{X}| \leqslant P_{\mathcal{G}} \log(6 k_2 L L_{\mathcal{G}}/\epsilon)$. Then, by a union bound over all $g$, whose parameter belongs to $\mathcal{X}$, we have:

$$
\mathbb{P}_{\mathcal{S}} \left[ \sup_{g \in \mathcal{X}} \left| \mathbb{E}_{\mathcal{S}} \sup_{f \in \mathcal{F}} \left[ \mathop{\mathbb{E}}_{\mathbf{x} \sim \hat{\mathcal{D}}} f(\mathbf{x}) - \mathop{\mathbb{E}}_{\mathbf{z} \sim \mathcal{N}} f \circ g(\mathbf{z}) \right] - \sup_{f \in \mathcal{F}} \left[ \mathop{\mathbb{E}}_{\mathbf{x} \sim \hat{\mathcal{D}}} f(\mathbf{x}) - \mathop{\mathbb{E}}_{\mathbf{z} \sim \mathcal{N}} f \circ g(\mathbf{z}) \right] \right| \geqslant \frac{\epsilon}{2} \right]
$$

$$
\leqslant 2 \exp(P_{\mathcal{G}} \log(6 k_2 L L_{\mathcal{G}}/\epsilon)) \exp(-\frac{\epsilon^2/2}{2/m}).
$$

Take $\epsilon = \frac{c_\delta \sqrt{2 P_{\mathcal{G}} \log(6 k_2 L L_{\mathcal{G}})}}{\sqrt{m}}$, then the proof is finished. □

*Proof of theorem 4.1.*

$$\sup_{g \in \mathcal{G}} |d_{\mathcal{F}}(\mathcal{D}_{real}, \mathcal{D}_g) - d_{\mathcal{F}}(\hat{\mathcal{D}}_{real}, \mathcal{D}_g)|$$

$$\leqslant \left| d_{\mathcal{F}}(\mathcal{D}_{real}, \mathcal{D}_g) - \mathbb{E}_{\mathcal{S}}[d_{\mathcal{F}}(\hat{\mathcal{D}}_{real}, \mathcal{D}_g)] \right| + \left| \mathbb{E}_{\mathcal{S}}[d_{\mathcal{F}}(\hat{\mathcal{D}}_{real}, \mathcal{D}_g)] - [d_{\mathcal{F}}(\hat{\mathcal{D}}_{real}, \mathcal{D}_g)] \right|$$

We complete our ultimate proof by combining lemma A.3 and A.4. □

## B  DETAILED EXPERIMENTAL SETTINGS

### B.1  DATA SET.

The training data set $\mathcal{S}^{train}$ is a point cloud in $\mathbb{R}^2$. We generate a data sample $\mathcal{S}$ accordingly:

1. Set the size of three Gaussian mixtures as $\{m_i\}_{i=1}^3$, $(m_1, m_2, m_3) \sim PN(m : \frac{1}{3}, \frac{1}{3}, \frac{1}{3})$.
2. Generate three Gaussian mixtures, respectively. $\mathcal{S} = \mathcal{S}_1 \cup \mathcal{S}_2 \cup \mathcal{S}_3$, where $\forall i, \mathcal{S}_i$ denotes $\{\{\mathbf{x}_j\}_{j=1}^{m_i} | \{\mathbf{x}_j\}_{j=1}^{m_i} \overset{i.i.d}{\sim} N_2(\boldsymbol{\mu}_i, \boldsymbol{\Sigma}_i), \forall j, \|\mathbf{x}_j\|_2 \leqslant 10\}$. We constrain the training data with $\forall j, \|\mathbf{x}_j\|_2 \leqslant 10$ to simulate the assumption of $\mathcal{S}$ without rejecting too much generated data.

According to the previous discussions, the generalization error $|d_{\mathcal{F}}(\mathcal{D}_{real}, \mathcal{D}_g) - d_{\mathcal{F}}(\hat{\mathcal{D}}_{real}, \mathcal{D}_g)|$ depends on $\mathcal{D}_{real}$, which is an unknown distribution. In order to compute the generalization error by an empirical approach, we generate two data sets: $\mathcal{S}^{train}, \mathcal{S}^{test}$, with $\#\mathcal{S}^{train} \ll \#\mathcal{S}^{test}$. In other words, we regard the uniform distribution over $\mathcal{S}^{test}$ as an approximation of $\mathcal{D}_{real}$. More concretely, we generate $\mathcal{S}^{train}$ with $m_{train} = 2 \times 10^4$ and $\mathcal{S}^{test}$ with $m_{test} = 2 \times 10^5$. We set $\boldsymbol{\mu}_1^\top = (-0.5, -1), \boldsymbol{\mu}_2^\top = (0.2, 0), \boldsymbol{\mu}_3^\top = (-1, 0), \forall \boldsymbol{\Sigma}_i = 0.05 \cdot \mathbf{I}_2$. We generate noise as $\mathbf{z} \sim N_2(\mathbf{0}, \mathbf{I})$ in the training process.

### B.2  NETWORK STRUCTURE AND PARAMETERS SETTING.

The structure of the discriminator is a three-layer$(2 \times 50$ FC)-ReLU-$(50 \times 50$ FC)-ReLU-$(50 \times 50$ FC)-ReLU-$(50 \times 1$ FC) network, where FC denotes a fully connected layer. The generator is a three-layer$(2 \times d$ FC)-ReLU-$(d \times 50$ FC)-ReLU-$(50 \times 50$ FC)-ReLU-$(50 \times 2$ FC) network, where $d$ takes value in $\{50, 70, 90, 110, 130\}$. For the generator, we apply the Xavier initialization with $gain = 1.3$ to the weights, and we set the initial bias as $0.1$. For the discriminator, we fill the weights with random numbers from $N(0, 0.02^2)$ and bias with $0$.

We use the optimizer of Adam Algorithm Kingma & Ba (2014) with $\beta = (0, 0.9)$ for both the discriminator and generator. The leaning rate of both the discriminator and generator are set to 8e-4. To apply $\ell_{2,2}$ weight normalization to the discriminator, we add a $\ell_{2,2}$ penalty, $(\prod_{i=1}^4 \|\mathbf{M}_{f,i}\|_{2,2} - 1)^2$, to the loss function of WGANs. We train 150 epochs with a batch size of 1000 to train a generator. Since there is no sufficient stopping criterion , we stop the training when the generated point cloud visually resembles Strain without further variance.

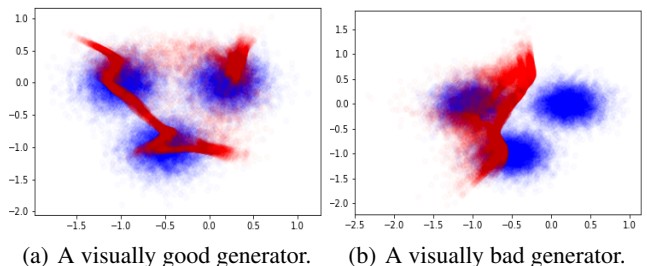

(a) A visually good generator.  (b) A visually bad generator.

Figure 3: The blue points cloud represents $\mathcal{S}^{train}$, a sample from a Gaussian Mixture distribution. The red points are $\{g(\mathbf{z}_i)\}_{i=1}^{m_{train}}$, while $\{\mathbf{z}_i\}_{i=1}^{m_{train}} \overset{i.i.d}{\sim} N_2(\mathbf{0}, \mathbf{I})$.

The process for computing $d_{\mathcal{F}}(\widehat{\mathcal{D}_{real}}, \mathcal{D}_g), d_{\mathcal{F}}(\widehat{\hat{\mathcal{D}}_{real}}, \mathcal{D}_g)$ is executed as follows:

1. Train a WGAN over $\mathcal{S}^{train}$ and save the parameters of the optimal $\bar{g}$.

2. Train a discriminator $f$ to maximize $\mathbb{E}_{\mathbf{x}\sim\hat{\mathcal{D}}_{train}}[f(\mathbf{x})] - \mathbb{E}_{\mathbf{x}\sim\mathcal{D}_{\bar{g}}}[f(\mathbf{x})]$ to obtain $\bar{f}_{train}$.

3. Train a discriminator $f$ to maximize $\mathbb{E}_{\mathbf{x}\sim\hat{\mathcal{D}}_{test}}[f(\mathbf{x})] - \mathbb{E}_{\mathbf{x}\sim\mathcal{D}_{\bar{g}}}[f(\mathbf{x})]$ to obtain $\bar{f}_{test}$.

4. We calculate $d_{\mathcal{F}}(\widehat{\hat{\mathcal{D}}_{train}, \mathcal{D}_g})$ by following section 3.1.1. We obtain $d_{\mathcal{F}}(\widehat{\mathcal{D}_{real}, \mathcal{D}_g})$ by substituting '$train$' with '$test$'.

## C   A NOVEL HYPOTHESIS TEST FOR $H_0 : \mathcal{D}_{real} = \mathcal{D}_g$.

For a given generator $g \in \mathcal{G}$, we are insterested in whether this generator generates real data. To answer this question, we formulate a hypothesis test.

Since our goal is to provide a hypothesis-testing process to figure out whether $\mathcal{D}_g = \mathcal{D}_{real}$ holds, at a confidential level $1 - \alpha$, we state the null hypothesis as $H_0 : \mathcal{D}_g = \mathcal{D}_{real}$ and the alternative hypothesis as $H_1 : \mathcal{D}_g \neq \mathcal{D}_{real}$. With the distribution distance, we revise the null hypothesis as: $H_0 : d_{\mathcal{F}}(\mathcal{D}_{real}, \mathcal{D}_g) = 0$. Intuitively, in the context of image-generating GANs, a visually bad generator should give evidence to reject the null hypothesis and embrace the alternative hypothesis.

Assume that $H_0$ holds, we have $d_{\mathcal{F}}(\hat{\mathcal{D}}_{real}, \mathcal{D}_g) \leqslant |d_{\mathcal{F}}(\hat{\mathcal{D}}_{real}, \mathcal{D}_g) - d_{\mathcal{F}}(\mathcal{D}_{real}, \mathcal{D}_g)|$. According to corollary 3.5, for a given $g \in \mathcal{G}$, with probability at least $1 - \alpha$ over the choice of $\mathcal{S}$, it holds that: $d_{\mathcal{F}}(\hat{\mathcal{D}}_{real}, \mathcal{D}_g) \leqslant T$, where $T$ denotes the right hand side of corollary 3.5. Hence, $d_{\mathcal{F}}(\hat{\mathcal{D}}_{real}, \mathcal{D}_g) > T$ provides significant evidence to reject $H_0$ at a $1 - \alpha$ confidential level. In other words, for a given $g \in \mathcal{G}$, if $d_{\mathcal{F}}(\hat{\mathcal{D}}_{real}, \mathcal{D}_g) > T$ holds. This means that the artificial data is discriminable and $g$ fails to adequately fit the real data. Hence, to make a hypothesis testing conclusion, we compare the values of $d_{\mathcal{F}}(\hat{\mathcal{D}}_{real}, \mathcal{D}_g)$ and $T$.

### C.0.1   HYPOTHESIS TESTING PROCESS.

1. For a given $g \in \mathcal{G}$, we train a discriminator $f$ to maximize $\mathbb{E}_{\mathbf{x}\sim\hat{\mathcal{D}}_{real}}[f(\mathbf{x})] - \mathbb{E}_{\mathbf{x}\sim\mathcal{D}_g}[f(\mathbf{x})]$. Notice that, training such a discriminator is equivalent to training a WGAN with $g$ fixed. Hence, we obtain $f_{train}$.

2. For $i \in \{1, ..., 50\}$, we have $d_{\mathcal{F}}(\widehat{\mathcal{D}_{real}, \mathcal{D}_g})_i = \mathbb{E}_{\mathbf{x}\sim\hat{\mathcal{D}}_{real}}[f_{train}(\mathbf{x})] - \mathbb{E}_{\mathbf{x}\sim\mathcal{D}_g}[f_{train}(\mathbf{x})]$. In each iterations, we generate an $m$-size noise sample set from $\mathcal{N}_{l_0}(\mathbf{0}, \mathbf{I})$, to feed the second term. Then, let $d_{\mathcal{F}}(\widehat{\mathcal{D}_{real}, \mathcal{D}_g}) = \sum_{i=1}^{50} d_{\mathcal{F}}(\widehat{\mathcal{D}_{real}, \mathcal{D}_g})_i / 50$.

3. We set $\hat{c}_{f,i}$ as $\|\mathbf{M}_{f_{train},i}\|_{p,q}$. For a given confidential level $1 - \alpha$, by plugging $\{\hat{c}_{f,i}\}_{i=1}^{k_1}$ into the right hand side of corollary 3.5, we obtain $\hat{T}$.

4. If $d_{\mathcal{F}}(\widehat{\mathcal{D}_{real}, \mathcal{D}_g}) > \hat{T}$, we reject $H_0$ at a $1 - \alpha$ level.

The hypothesis testing for $H_0$ is an application of corollary 3.5. Instead of providing a statistic with an explicit distribution, our hypothesis testing is based on a probabilistic bound for generalization error with a fixed generator.

Notice that, there are some distribution-free methods to compare two distributions in a purely statistical way, such as the Kolmogorov-Smirnov test, Shapiro-Wilk test, Mann-Whitney U Test and Bootstrap Methods. However, most of these methods are heavily based on some strong assumptions (i.e., normality, the conditions for central limit theorem and low dimension). Hence, these methods do not work with some unusual and high dimensional distributions. In high dimensional cases, corollary 3.5 guarantees that, our novel hypothesis testing has the capacity to discriminate between high dimensional distributions. Our numerical experiments show that, this hypothesis testing is commensurate with our observations.

## D   MORE EXPERIMENTAL RESULTS FOR OUR HYPOTHESIS TEST

The following experiment illustrates the hypothesis testing process in section 3.1.1. For a specific generator, we convey the hypothesis process on 100 sample sets $\{\mathcal{S}^i\}_{i=1}^{100}$, where $\forall \mathcal{S}^i$ is derived from

the same distribution of $\mathcal{S}^{train}$. If a generator is visually bad, the hypothesis process is expected to reject $H_0$ almost 100 times, at a 0.95 confidential level. In these cases, it is highly probable that the generated distribution is not close to the real distribution. Notice that, even if the hypothesis process embraces $H_0$, the generator is not guaranteed to fit the real distribution well. Instead, we should say, the sample data is not rich enough to provide significant evidence to reject $H_0$ at a high confidential level. In fact, for a visually good generator, the hypothesis process fails to reject $H_0$, even with a rich sample data set.

### D.0.1 Hypothesis testing for $H_0 : \mathcal{D}_{real} = \mathcal{D}_g$.

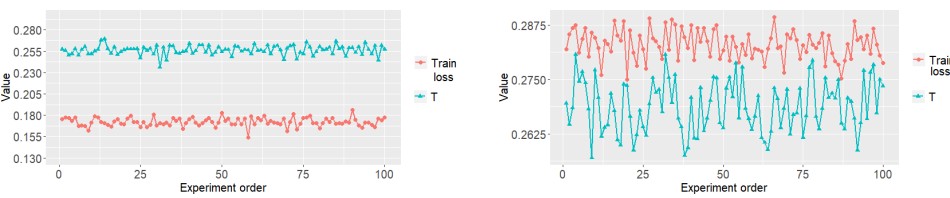

(a) For the previous visually good generator.  (b) For the previous visually bad generator.

Figure 4: A representative result of the hypothesis test with the generators in figure 3. In every experiment, we reject $H_0$ if the training loss is larger than T.

We show a representative result in figure 4. Recall that, the hypothesis testing process reject $H_0$ if $d_{\mathcal{F}}(\widehat{\mathcal{D}_{real}}, \mathcal{D}_g) > \hat{T}$, where we refer to $d_{\mathcal{F}}(\widehat{\mathcal{D}_{real}}, \mathcal{D}_g)$ as the training loss and $\hat{T}$ as the test statistic for $H_0$. Figure 4 suggests that, the hypothesis testing process embraces $H_0$ with a visually good generator, while it reject $H_0$ with a visually bad one, at a 0.95 confidential level. In fact, our experiments show that the conclusions from the hypothesis testing are commensurate with our observation. In our toy model, $H_0$ is rejected when the generated data forms a single clique (what we call Model Collapse) or it obviously disjoints with the real data. Hence, our hypothesis test is capable of discriminating between good and bad generators, with sufficient sample data.

In our experiments, an extremely small and negative value of gap $d_{\mathcal{F}}(\widehat{\mathcal{D}_{real}}, \mathcal{D}_g) - \hat{T}$ implies that the generator fits the real data very well and vice versa. For the previous bad generator in figure 3, the gap is positive but small. This implies that, though the generator fails to fit the real data, it does not violate the real data grossly.

We display some representative results for hypothesis test in section 4. The experimental settings are the same as in previous discussions.

### D.1 Discrimination

To verify that our hypothesis test shows discrimination in choice of good generators, we conduct hypothesis test on 79 generators with the same training set $\mathcal{S}_{train}$ and $d = 50$.

In figure 6 and 7, we juxtapose the hypothesis test conclusion and the corresponding generator. Recall that, for a given $g$, we reject $H_0 : \mathcal{D}_{real} = \mathcal{D}_g$ if $d_{\mathcal{F}}(\widehat{\mathcal{D}_{real}}, \mathcal{D}_g) > \hat{T}$. Numerical experiments show that, our hypothesis is capable to detect the visually bad generators, at a 0.95 confidential level.

Our hypothesis test is based on a probabilistic inequality, which depends on the artificially designed sample set $\mathcal{S}_{train}$ and confidential level $1 - \alpha$. Thus, for visually bad generators, there may be some rare acceptances of $H_0$. In these cases, instead of stating 'the hypothesis test provided an incorrect conclusion', we should say 'the sample set $\mathcal{S}_{train}$ is not rich enough to provide evidence to reject $H_0$, at a $1 - \alpha$ confidential level'. Such an example is included in 7.

### D.2 Distribution deviation sensitivity

The numerical experiments also imply that, the hypothesis test can detect how $\mathcal{D}_g$ violates $\mathcal{D}_{real}$. We provide an example in figure 5.

According to figure 5, the hypothesis test provides stable rejection of $H_0$ with $\mathcal{S}_i$, $i = 1, \ldots, 100$. Moreover, there is an evident gap between the training loss and T. Recall that the gap between the training loss and T of the bad generator in figure 3 is small and unstable. Obviously, the generator in figure 5 is much worse than the previous ones. The result suggests that, our hypothesis test is sensitive to the deviation between $\mathcal{D}_g, \mathcal{D}_{real}$.

Concretely, for generators that fit the data very well (or very badly), the hypothesis test provides a training loss which is much larger (or lower) than $T$; for generators that perform modestly, the hypothesis test provides a training loss that is close to $T$ to show its hesitation. In latter cases, to obtain an unambiguous conclusion, we can choose a richer sample set or set a larger confidential level.

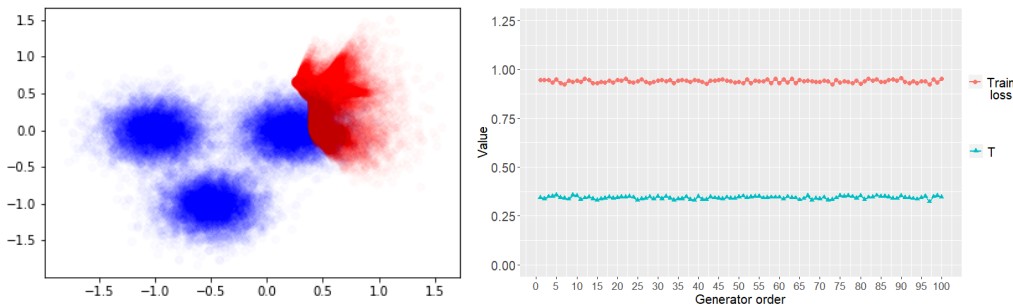

(a) An extremely bad generator, which is even worse than figure 3. Model collapse also occurs.

(b) We convey the hypothesis test with $\mathcal{S}_i$, $i = 1, \ldots, 100$. All the experiments reject $H_0$ at a 0.95 confidential level.

Figure 5: In such a extreme case, the values of training loss and $T$ are stable.

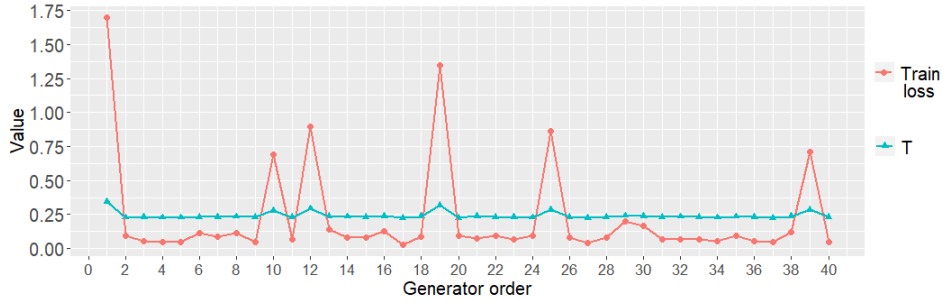

(a) The hypothesis test result. For a fixed $g$, we reject $H_0$ if the training loss is larger than T. According to the figure, we reject $H_0$ with the $1, 10, 12, 19, 25, 39$ th generators.

(b) 39 generator examples in order. Notice that generators $1, 10, 12, 19, 25$ and $39$ are visually bad (or even occur model collapse), which is commensurate with our hypothesis test conclusion.

Figure 6: In every figure, the blue points cloud represents $\mathcal{S}^{train}$, a sample from a Gaussian Mixture distribution. The red points are $\{g(\mathbf{z}_i)\}_{i=1}^{m_{train}}$, while $\{\mathbf{z}_i\}_{i=1}^{m_{train}} \overset{i.i.d}{\sim} N_2(\mathbf{0}, \mathbf{I})$.

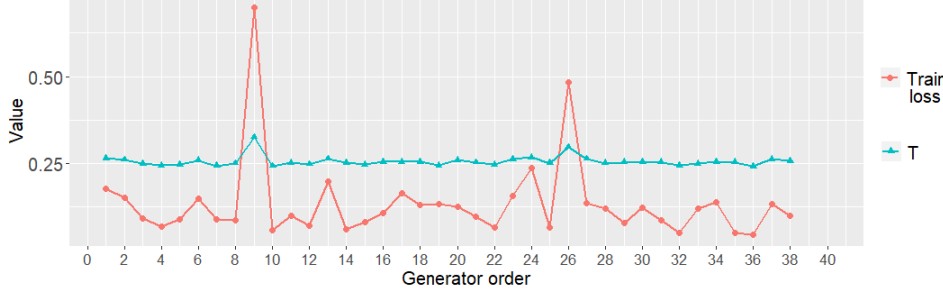

(a) The hypothesis test result. For a fixed $g$, we reject $H_0$ if the training loss is larger than T. According to the figure, we reject $H_0$ with the $9, 26$ th generators.

(b) 40 generator examples in order. Notice that generator $9, 24$ and $26$ are visually bad (or even occur model collapse).

Figure 7: Though our hypothesis test fails to detect the 24 th generator, it shows that train loss is extremely closed to T for the 24 th generator. This unusual acceptance is caused by randomness. In this case, the sample set $\mathcal{S}_{train}$ is not rich enough to reject $H_0$, at a 0.95 confidential level.

