# OpenReview forum: "A Uniform Generalization Error Bound for Generative Adversarial Networks"
_ICLR.cc/2020/Conference — Reject_

### Official Review · AnonReviewer1 · 2019-10-09
**Official Blind Review #1**

**Rating:** 3

**Review:**


The authors give new generalization bounds for GANs.  The argue for a new definition of generalization for GANs, which isolates the effect of sampling error arising from sampling from
the real distribution.  (Their argument, which I find convincing, may be paraphrased as saying that sampling from the generator should be viewed more as a computational cost than a data-gathering cost, since a procedure may sample from a generator as many times as it wants.)  They give a bound that is uniform both over discriminators and generators.

In my opinion, the mathematical writing is not up to the standard for publication in ICLR.  There are many cases where the paper is unclear, and, in many other cases, I have to guess what they mean, and I have limited confidence in my guess.  Since the theorems of the paper are quite technical, it is very difficult to be confident what their exact statements are.  One example is where they write
"We define F with weight normalization as ...".  The fact that the c in this definition has a subscript of f led me to think that the bound can depend on f, but this does not make sense.  I assume that the bound is independent of f.   Later, when they write "we define the Lipschitz constant of f", since they write that this Lipschitz constant is with respect to a norm on parameterizations, I take it that they are defining the Lipschitz constant of a mapping from parameters to functions (and not a Lipschitz constant of f).  But they don't indicate what metric is used to define the distance between functions.

When the compare their bound with previous work, they treat quantities as constants which can be large.  For example, they treat the product of the operator norms of the layers as a constant.  It also is not clear how they get the bound that they attribute to Bartlett, et al from the bound in that paper.

Most of the technical heavy lifting appears to have been borrowed from the Chen, et al paper.

A more detailed account of how they get their covering bound for their (p,q) norm from the
Dumer paper is needed.

Since the authors assume that phi is the identity, it seems unnecessary to keep subscript d with it.  On the other hand, subscripting d with calF would make the theorems easier to interpret at a glance.

(Edit on 11/14/19: I have read the response and checked the revision.  Some of the above criticisms have been addressed by the revision.  I have increased my rating.)

**Experience Assessment:**

I have published one or two papers in this area.

**Review Assessment: Checking Correctness Of Derivations And Theory:**

I assessed the sensibility of the derivations and theory.

**Review Assessment: Checking Correctness Of Experiments:**

I assessed the sensibility of the experiments.

**Review Assessment: Thoroughness In Paper Reading:**

I read the paper at least twice and used my best judgement in assessing the paper.

---

> ### Author Response · Authors · 2019-11-13
> **Response to Reviewer #1**
>
> 1. The logical structure and exact statements
>
> In summary, our main statements are concluded in section 3, 4. Our statements show that, the Complexity of F is proportional to generalization error bound and the complexity of G is in reverse proportion to the probability, of which the bound holds. Hence, our theory offers us a guidance to reduce the Complexity of F,G and hence improve the robustness of GANs. According to the corollary 4.2, 4.3, such guidance on parameter designs to train robust GANs can be concrete, which is poorly studied in previous works: For ell_{p,q} normalization, we can set 1/p+1/q >= 1 and the production of the upper bound of the ell_{p,q} norm of weight matrices, c_{f,i}, c_{g,i}, as small constants. For spectral normalization, we can set the production of the upper bound of the spectral norm of weight matrices, B_{F,i}, B_{G,i}, as small constants.
>
> 2. The definition of the Lipschitz constant of f:
>
> Thanks for your attention. We do apologize for my misuse of notations. In this paper, we denote f_w as the mapping from the parameter space to the output space, that is [-1,1] (the range of discriminator function f is at the 2nd paragraph in Section 2). Hence, we did not set the range as a function space and we adopted Euclid norm for [-1,1] in R.
>
> The subscript f of c_{f,i} should be substituted by c_{\mathcal{F},i}. We set this subscript just for distinguishing between the bound we set for F and G. In fact, the upper bound of weight normalization only depends on the setting of the network class F, G, which is initially determined. Hence, our generalization bound is independent of f. It is commensurate with intuition.
> We will add these remarks after the definitions as soon as possible.
>
> 3. The production of c_{f,i}(operator norms) can be artificially constrained to a small constant:
>
> Our paper established explicit formulation for generalization bound with a concrete network parameter setting (we adopt ell_{p,q} weight normalization). Since the ell_{p,q} norm bounds of weight matrices are artificially designed, we can constrain the bound to small constants before the training process. In other words, we can take advantage of ell_{p,q} normalization to obtain a parameter-depend generalization bound, which is be tighter than previous works, by setting small ell_{p,q} norm bounds for weight matrices. It can also explain why such normalizations can improve the generalization capacity of GANs.
>
> 4. The novelty of our paper:
>
> Our main result, theorem 4.1, shows our novelty: we explain how the complexity of F, G contribute to the generalization bound. The result of the Rademacher Complexity in Chen et al is merely established under the regression problems. We adopt their result to obtain the estimation of the complexity of F. However, the novelty of our paper is exploring the interaction between the complexity of F and G for GANs. As far as I know, we are the first to formulate a uniform generalization error bound. In the theorem 4.1, the most essential point is, how the epsilon term determines the probability of which the bound holds and how that probability depend on the complexity of G.
>
> Hence, though the formulation we borrowed from Chen et al seems long, it was not the essential part. Instead, we should focus on the epsilon term in theorem 4.1. To avoid misunderstanding, we may substitute epsilon with other notations.
>
> 5. The proof of Covering Number:
>
> We apologize for our omission in the proof. We will add necessary proof details about how we obtain the covering number. In fact, by adopting \ell_{p,q} normalization, we can obtain a better estimation of the covering number of G. while the covering number of Arora et al’s is O(P_G log(P_G Const)), our result can be O(P_G log(k_2 Const)).
>
> 6. For the notation d_\phi(,):
>
> Your advice is greatly helpful. We would substitute the unnecessary subscript \phi with cal F for a laconic and more interpretable form.

---

> > ### Comment · AnonReviewer1 · 2019-11-14
> > **read your response**
> >
> > I have read your response and noted that you provided a proof of the covering bound in your revision, and clarified the definitions.
> >
> > I'm still not convinced by your justification for viewing the product of the bounds on the norms as a constant.  I agree that these can be chosen to be small, but if they are too small, then the network is not expressive enough to be useful.
> >
> > I also took note of your other comments about what is novel, etc. , though I still don't see substantial technical advance in this work.
> >
> > I have raised my rating.

---

### Official Review · AnonReviewer2 · 2019-10-23
**Official Blind Review #2**

**Rating:** 3

**Review:**

In this paper, the authors study the generalization bound for GANs based on a new definition of generalization error where the distribution corresponding to the generator is assumed to be known for each generator (i.e., there is no empirical distribution for generators). For this generalization error, the authors give both bounds for a fixed generator and a uniform bound for a class of generators.

In my opinion, most of the theoretical results seem follow directly from standard tools in statistical learning theory and existing results on capacity bounds of neural networks. It seems that the authors do not introduce new ideas or techniques in the analysis.

The authors made comparisons with the related results in Arora et al (2017). However, since the generalization bound in Arora et al (2017) is based on a different definition of generalization error where empirical distributions are considered for both discriminators and generators, the comparison seems not fair.

----------------------
After rebuttal:

I have read the authors' response. The contribution seems not novel and enough. I would like to keep my original score.

**Experience Assessment:**

I have read many papers in this area.

**Review Assessment: Checking Correctness Of Derivations And Theory:**

I assessed the sensibility of the derivations and theory.

**Review Assessment: Checking Correctness Of Experiments:**

I did not assess the experiments.

**Review Assessment: Thoroughness In Paper Reading:**

I read the paper at least twice and used my best judgement in assessing the paper.

---

> ### Author Response · Authors · 2019-11-13
> **Response to Reviewer #2**
>
> Thank you for the detailed reviews. We have updated the manuscript according to your suggestions and answer specific questions below.
>
> 1. The novelty of our paper.
>
> Our main result, theorem 4.1, shows our novelty: we explain how the complexity of F, G contribute to the generalization bound. The result of the Rademacher Complexity in Chen et al is merely studied under the regression problems. We adopt their result to obtain the estimation of the complexity of F. However, our paper novelty explores the interaction between the complexity of F and G for GANs. And we are the first to formulate a uniform generalization error bound. In the theorem 4.1, the most essential point is, how the epsilon term determines the probability of which the bound holds and how that probability depend on the complexity of G. Hence, though the formulation we borrowed from Chen et al is seeming long, it was not the essential part. Instead, we should focus on the epsilon term in theorem 4.1. To avoid misunderstanding, we may substitute epsilon with other notations.
>
> Beyond applying union bound, in our proof process, we take advantage of the \ell_{p,q} weight normalization to obtain a better upper bound of the covering number. The covering number of Arora’s work is O(P_G log(P_G Constant)), while our result can be O(P_G log(k_2 Constant)). (k_2: the depth of G, P_G: the parameter number of G). We will fulfill this blank as soon as possible. Moreover, we established explicit formulation of generalization bound with network parameters. Our generalization bound provides guidance on parameter designs for training robust GANs.
>
>
> 2.	The comparisons with Arora et al
>
> In fact, by adopting our definition of generalization error and the technique in Arora et al, we can still proof that, our generalization error bound (with g fixed) is still tighter than Arora et al. The comparisons will be updated to the manuscript as soon as possible.
>
> Concretely, according the proof of Theorem B.2 of Arora et al, if we adopt our definition of generalization error, the formula (6) no longer contribute to the generalization bound. Hence, the RHS of Theorem 3.1 becomes epsilon/2, which does not impact our previous comparisons.

---

### Official Review · AnonReviewer3 · 2019-10-30
**Official Blind Review #3**

**Rating:** 1

**Review:**

This paper proves generalization bounds for GANs. I think the paper can be improved significantly in several ways:

1- Writing: The first two sections are relatively well-written. The problem starts at section 3 and continues after that. Some of the things that can be improved:

a) The discussion on the different definitions of generalizations is not really helpful in the current format. You might want to explain how these different definitions relate to each other. For example, if generalization in one of them implies generalization in the other one, etc.

b)Theorem 2.3 is a general statement but it is followed by Corollary 3.3 which is a very specific generalization bound. There is no explanation how one can show the corollary.  Even worse is mixing these two in the proof of the theorem in the appendix.   Please consider improving the use of Theorems, Lemmas and Corollaries.

c) Section 3 and 4 have bunch of theorems and collieries without much explanation. It is not clear that all of these are actually helpful for the main purpose of the paper.

d) I don't completely understand the notation in Corollary 3.3. Eg. what is d_{f,\ell}?

2) Related Work: I think authors need to do a more comprehensive literature review on generalization bounds. Since the generalization bounds presented here are built on the supervised learning bounds, authors discuss the generalization bounds in supervised learning. For example, authors heavily rely on Chen et al. (2019) for their generalization bounds while very similar results where shown before by [1] and [2].
[1] Neyshabur, Behnam, Ryota Tomioka, and Nathan Srebro. "Norm-based capacity control in neural networks." Conference on Learning Theory. 2015.
[2] Golowich, Noah, Alexander Rakhlin, and Ohad Shamir. "Size-independent sample complexity of neural networks." Conference on Learning Theory. 2018.

3) Definition of generalization: I don't think the definition of generalization suggested in this work is much different than Arora et. al. since f really doesn't depend on samples from D_g and hence the empirical and true distributions are not very different. In fact, I think the definition provided by Arora et. al. 2017 is preferred because at the end of the day, we have to estimate the distribution D_g by generating some samples.

4) Generalization bound for fixed g: Unfortunately, the novelty of these generalization bounds are very limited as they are a direct application of known generalization bounds in the supervised settings. Therefore, the authors contributions are very limited here.

5) Generalization bounds for all generators: Again, here the novelty and final result is very limited since the bounds achieved by a union bound arguments and does not really go beyond that.

6) Experiments: Experiments can also be improved significantly. Currently, the correlation is reported for 5 trained networks and it is not clear to me that this result is statistically significant. Moreover, only one hyper-parameter is changed in the experiments which could be problematic. I suggest authors to change multiple hyper-parameters and train more networks to improve the evaluation.


******************************

After author rebuttals:

I have read the authors response and looked at the revision. Unfortunately, many of my concerns are not addressed adequately so my score remains the same.

**Experience Assessment:**

I have published in this field for several years.

**Review Assessment: Checking Correctness Of Derivations And Theory:**

I assessed the sensibility of the derivations and theory.

**Review Assessment: Checking Correctness Of Experiments:**

I assessed the sensibility of the experiments.

**Review Assessment: Thoroughness In Paper Reading:**

I read the paper at least twice and used my best judgement in assessing the paper.

---

> ### Author Response · Authors · 2019-11-13
> **Response to Reviewer #3**
>
> Thank you for the detailed reviews. We have updated the manuscript according to your suggestions and answer specific questions below.
>
> 1. The interpretation of our theorems
>
> In section 3, we established a generalization bound with a fixed generator in theorem 3.2 in a general version. Then, we showed the formulation of such a bound in Corollary 3.3, 3.5 and 3.6, by adopting \ell_{p,q} normalization and Spectral normalization. Our main statement for section 3 is at the second line after theorem 3.2: our generalization bound shows that the generalization bound relies on the Complexity of F. It states that we can reduce the generalization bound by reducing the Complexity of F and hence control the generalization error of GANs.
>
> At the beginning of Section 4, our analysis shows, the previous bound does not hold for every g. Our main result is the successive theorem 4.1, which provides a uniform bound. Notice that, the epsilon term in theorem 4.1 is not a neglectable constant. Instead, it relies on the complexity of G and determines the probability of which the bound holds. Interpretations follow the theorem 4.1 shows, a grossly Complexity of G would reduce the probability of which the bound hold. Our main result shows we should reduce the Complexity of G in a reasonable range, hence, we can make the bound holds at a higher confidential level. In Corollary 4.2, 4.3, we provide a explicit formulation for such a uniform bound, by adopting ell_{p,q} normalization and Spectral normalization.
>
>
> 2. The relationship between different definitions of generalization error
>
> We hope to establish a definition to depict the gap between the ‘training error’(empirical distance) and the ‘testing error’(popular distance) in our experiment process.
>
> However, our discussions showed that, the two previous definitions (Arora et. al, Zhang et. al) are not commensurate with the experiment process and the three different definitions are not equivalent.
>
> Arora et al define the generalization error as |d(\hat{D}_g, \hat{D}_{real})- d({D}_g, {D}_{real})|, where \hat{D}_g stands for the uniform distribution over the generated noise set. Assumed that our training set (real data set) is S, with #S=m. In every epoch, we generated m new Gaussian noise as input of G. Hence, if we train the GAN for n epoch, then our \hat{D}_g is defined over a noise data set S_noise of size m*n. However, d(\hat{D}_g, \hat{D}_{real}) means that, the m*n noise set is fixed before the whole training process. In every epoch, we select a subset of size m randomly from S_noise (NOT from Gaussian distribution). It is possible that, an element z\in S_noise is selected twice in two epochs. However, according to the training process, every Gaussian noise vector is used only once. Arora et al’s definition faultily treat the generated noise as a static ‘training set’, which is not commensurate with the practical process.
>
> 3. We do apologize for our omission of some proof details and some potential typos. We will improve the proof as soon as possible. In fact, d_{f,\ell} merely stands for d_{\ell}, that is, the dimension of the \ell th layer of f.
>
> 4. The novelty of our paper:
>
> The result of the Rademacher Complexity in Chen et al is studied under the regression problems. We adopt their result merely to obtain the estimation of the complexity of F. Our main result, theorem 4.2, shows our novelty. We explore the interaction between the complexity of F and G for GANs. And we are the first to formulate a uniform generalization error bound.
>
> In the theorem 4.2, the most essential point is, how the epsilon term determines the probability of which the bound holds and how that probability depend on the complexity of G. Hence, though the formulation we borrowed from Chen et al is seeming long, it was not the essential part. Instead, we should focus on the epsilon term in theorem 4.2. To avoid misunderstanding, we may substitute epsilon with other notations.
>
> The papers [1] and [2] are clearly relevant. We have read these papers before and we would add the references to the paper. Though the result of [1] is similar to Chen et al, the bound of [1] relies on 2^(d-1), where d is the depth of neural networks. Hence, the result in Chen et al is tighter. Since [2] only provide result in Frobenius Norm, \ell_{2,1} and \ell_{1,\infty} norm, Chen et al is more general.
>
> Beyond applying union bound, in our proof process, we take advantage of the \ell_{p,q} weight normalization to obtain a better upper bound of the covering number. The covering number of Arora’s work is O(P_G log(P_G Constant)), while our result can be O(P_G log(k_2 Constant)). (k_2: the depth of G, P_G: the parameter number of G). We will update the definitions as soon as possible.
>
> 4.	More numerical experiments are under running.
> Thanks for your attention! More numerical experiments on varying multiple hyper-parameters are under running. We will update the new experiment data as soon as possible.

---

> > ### Comment · AnonReviewer3 · 2019-11-13
> > **Thanks for your response**
> >
> > Thanks for your response. The current revision is not very different than the original submission and my concerns about the writing, related work and experiments remain the same. I will take another look at the revision after the discussion period.

---

### Decision · Program_Chairs · 2019-12-19

**Decision:**

Reject

**Comment:**

The authors received reviews from true experts and these experts felt the paper was not up to the standards of ICLR.

Reviewer 3 and Reviewer 1 disagree as to whether the new notion of generalization error is appropriate. I think both cases can be defended. I think the authors should aim to sharpen their argument in this regard.Several reviewers at one point remark that the results follow from standard techniques: shouldn't this be the case? I believe the actual criticism being made is that the value of these new results do not go above and beyond existing ones. There is also the matter of what value should be attributed to technical developments on their own. On this matter, the reviewers seem to agree that the derivations lean heavily on prior work.